# GoldenStart: Q-Guided Priors and Entropy Control for Distilling Flow Policies

**He Zhang**[1,3], **Ying Sun**[1†], **Hui Xiong**[1,2†]

[1]Thrust of Artificial Intelligence, The Hong Kong University of Science and Technology (Guangzhou)
[2]Department of Computer Science and Engineering,
The Hong Kong University of Science and Technology, Hong Kong SAR
[3]AI[2] Robotics
`hzhang757@connect.hkust-gz.edu.cn`, `yings@hkust-gz.edu.cn`,
`xionghui@ust.hk`

## Abstract

Flow-matching policies hold great promise for reinforcement learning (RL) by capturing complex, multi-modal action distributions. However, their practical application is often hindered by prohibitive inference latency and ineffective online exploration. Although recent works have employed one-step distillation for fast inference, the structure of the initial noise distribution remains an overlooked factor that presents significant untapped potential. This overlooked factor, along with the challenge of controlling policy stochasticity, constitutes two critical areas for advancing distilled flow-matching policies. To overcome these limitations, we propose GoldenStart (GSFlow), a policy distillation method with Q-guided priors and explicit entropy control. Instead of initializing generation from uninformed noise, we introduce a Q-guided prior modeled by a conditional VAE. This state-conditioned prior repositions the starting points of the one-step generation process into high-Q regions, effectively providing a "golden start" that shortcuts the policy to promising actions. Furthermore, for effective online exploration, we enable our distilled actor to output a stochastic distribution instead of a deterministic point. This is governed by entropy regularization, allowing the policy to shift from pure exploitation to principled exploration. Our integrated framework demonstrates that by designing the generative startpoint and explicitly controlling policy entropy, it is possible to achieve efficient and exploratory policies, bridging the generative models and the practical actor-critic methods. We conduct extensive experiments on offline and online continuous control benchmarks, where our method significantly outperforms prior state-of-the-art approaches. Code will be available at `https://github.com/ZhHe11/GSFlow-RL`.

## 1 Introduction

Recent advances in policy learning have increasingly leveraged generative models to capture complex and multimodal policies (Chi et al., 2023; Ghugare & Eysenbach, 2025; Black et al., 2024a). Unlike traditional methods that assume a unimodal Gaussian distribution, these approaches model the rich action distributions required for sophisticated control tasks, demonstrating potential across a broader range of scenarios. However, this expressive power comes at a cost: The iterative nature of the generation process, which requires multiple steps to produce a single action, leads to prohibitive inference latency. This bottleneck makes such models impractical for real-time scenarios, such as Vision-Language-Action (VLA) models (Zhai et al., 2024; Black et al., 2025).

Flow matching has recently emerged as a more efficient alternative to diffusion models (Lipman et al., 2023; Liu et al.; Albergo & Vanden-Eijnden, 2023; Geng et al., 2025). This has spurred research into the acceleration of generative policies using flow matching (Braun et al., 2024; Agrawalla et al., 2025; Espinosa-Dice et al., 2025), although these approaches often still require multiple denoising steps at the inference stage. To address this, a more aggressive solution using

---

† Corresponding authors.

one-step distillation proves particularly effective by training a student network to emulate the entire multi-step transformation in a single forward pass (Park et al., 2025b). Although effective in reducing latency, these methods overlook two critical opportunities to improve policies.

First, their generative process begins from a fixed, uninformed prior, typically a standard Gaussian distribution. However, an emerging perspective in generative modeling suggests that initial noise is a critical component that can guide generation (Zhou et al., 2025; Ma et al., 2025b). We posit that an optimized starting point (a "golden start") can create a powerful learning shortcut to high-value actions. As illustrated in Figure 1, an informed prior (yellow) strategically shifted towards high-value regions provides a more direct path to optimal actions, compared to an uninformed Gaussian distribution (gray). The second opportunity stems from the deterministic mapping inherent in the distilled policies. Given a specific prior noise, the generator learns a "point-to-point" mapping, transforming a single noise vector into a single deterministic action. This architecture inherently lacks explicit control over policy stochasticity, which is crucial for effective online exploration (Ma et al., 2025a).

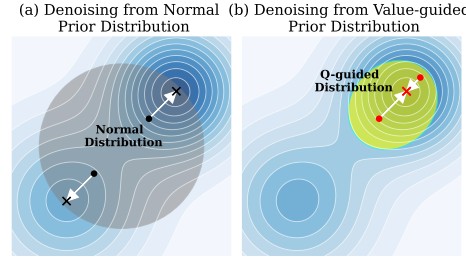

Figure 1: An illustration of denoising from an uninformed Gaussian prior (a) versus an informed, value-guided prior (b). Deeper blue indicates higher value.

To overcome these challenges, we introduce GoldenStart (GS-flow), a novel distillation framework that unifies high-speed inference with precise exploitation and adaptive exploration. Our work is built upon two key innovations: (1) First, we propose a Q-Guided Generative Prior, learned via a lightweight conditional VAE. This prior replaces the uninformative Gaussian noise with a state-aware distribution biased toward high-value actions, as identified by the critic. This provides the "golden start", effectively shortcutting the policy learning to optimal modes with negligible latency overhead. (2) Second, we introduce Entropy-Regularized Distillation, where the student policy learns a full distribution over actions, not just deterministic ones. This transforms the conventional "point-to-point" mapping into a more expressive "point-to-distribution" paradigm. During the online RL stage, an entropy regularization mechanism is activated, allowing the policy to dynamically modulate its stochasticity for robust exploration.

By co-optimizing the generative starting point and the output distribution, our framework improves the policy's ability to represent high-value actions while merging flow-based distillation models with adaptive exploration control. To this end, our approach, GS-flow, is extensively evaluated on continuous control benchmarks, including OGBench and D4RL (Park et al., 2025a; Fu et al., 2020). The results demonstrate that our method establishes a new state-of-the-art in overall performance. It particularly excels on complex tasks requiring multi-modal action representations and principled exploration, where it significantly outperforms prior methods.

## 2 PRELIMINARY

### 2.1 PROBLEM DEFINITION

A reinforcement learning problem is formulated as a Markov Decision Process (MDP) (Sutton et al., 1998), defined by the tuple $(S, A, P, r, \gamma)$. $S$ is the state space, $A$ is the action space, $P : S \times A \times S \to [0, 1]$ is the state transition probability function, $r : S \times A \to \mathbb{R}$ is the reward function, and $\gamma \in [0, 1)$ is the discount factor. A policy $\pi(a|s)$ is a distribution over actions given a state. The objective is to learn an optimal policy $\pi^*$ that maximizes the expected discounted cumulative reward, $J(\pi) = \mathbb{E}_{\tau \sim \pi} \left[ \sum_{t=0} \gamma^t r(s_t, a_t) \right]$, where $\tau = (s_0, a_0, s_1, a_1, \dots)$ is a trajectory sampled by executing the policy $\pi$. Offline RL involves learning from a static transition dataset $\mathcal{D} = \{(s_t, a_t, r_t, s_{t+1})\}_{t=1}^N$ without environmental interaction, where $N$ is number of steps in the dataset (Levine et al., 2020). The Offline-to-Online RL setting extends this problem by introducing a subsequent online interaction phase, also with the aim of maximizing the return function $J(\pi)$.

## 2.2 Distillation from Flow-Matching Policy

The significant inference cost of iterative flow-matching policies has motivated researchers to distill them into single-step, fast student policies (Park et al., 2025b). This approach, named FQL, operates within an actor-critic structure and trains the student actor with a hybrid objective: concurrently minimizing a distillation loss against the flow-matching teacher while maximizing the Q value. The framework utilizes two distinct models:

**Teacher Policy ($\pi_\phi$):** The flow-matching teacher policy is trained on the offline dataset $\mathcal{D}$ using a behavioral cloning (BC) objective. For a given state-action pair $(s, a)$ sampled from the dataset and a noise sample $x_0 \sim \mathcal{N}(0, I)$, the training objective is to learn a conditional velocity field $v_\phi(x_t, s, t)$. This field is parameterized by a time variable $t \in [0, 1]$ and defines a straight path between the noise $x_0$ and the action $a$ (Lipman et al., 2023; Liu et al.). Assuming $t$ is sampled uniformly from this interval ($t \sim U(0, 1)$), the interpolated action along this path is $x_t = (1 - t)x_0 + ta$. The Conditional Flow Matching (CFM) loss then trains the network to match the constant velocity of this path:

$$\mathcal{L}_{\text{CFM}}(\phi) = \mathbb{E}_{t \sim U(0,1),(s,a) \sim \mathcal{D}, x_0 \sim \mathcal{N}(0,I)} \left[ \|v_\phi(x_t, s, t) - (a - x_0)\|^2 \right] \tag{1}$$

During inference, the teacher policy $\pi_\phi$ generates a final action $a_{teacher}$ by using the trained $v_\phi$ to iteratively denoise an initial noise sample over multiple steps.

**Student Policy ($\pi_\varphi$):** The separate student network is trained for fast inference. It takes a state $s$ and a noise vector $x_0$ as input and produces an action in a single forward pass. The student policy $\pi_\varphi$ is trained to concurrently maximize the Q-value while staying close to the teacher's output. This is achieved by minimizing a compound loss function that combines a Q-learning objective with a distillation loss:

$$\mathcal{L}_{\text{FQL}}(\varphi) = \mathbb{E}_{s \sim \mathcal{D}, x_0 \sim \mathcal{N}(\mathbf{0}, \mathbf{I})} \left[ -Q(s, \pi_\varphi(s, x_0)) + \alpha \|\pi_\phi(s, x_0) - \pi_\varphi(s, x_0)\|^2 \right], \tag{2}$$

where $Q$ is the critic function learned within an actor-critic framework (Haarnoja et al., 2018) and the hyperparameter $\alpha$ controls the strength of the behavioral cloning (BC) regularization (Tarasov et al., 2023). In particular, $\pi_\varphi$ requires no iterative denoising at inference, as it is trained to directly approximate the multi-step denoising action in a single step.

## 2.3 Multi-Crescent Task

To demonstrate our insight, we design the Multi-Crescent environment, shown in Figure 2. The environment consists of six separate, nonconvex, crescent-shaped regions of high reward, designed to challenge agents that are prone to Q-value overestimation. The reward is structured into three levels: the top-left/bottom-right crescents provide a moderate reward, the middle-left/middle-right crescents provide a higher reward, and the globally optimal top-right/bottom-left crescents offer the maximum reward. All other areas yield zero reward. This setup emulates a complex environment with multiple levels of local optima.

When constructing the offline dataset, we deliberately excluded all samples from the two highest-reward crescent regions (top-left/bottom-right), as shown by the blue scatter points in Figure 5a. This environment poses two challenges to the algorithms: 1) During the offline learning phase: The algorithm needs to identify and converge to the higher-reward mode present within the dataset (middle-left/middle-right) while suppressing Q-value overestimation for unseen regions. 2) During the online exploration phase: The algorithm must demonstrate efficient exploration to discover and exploit the globally optimal modes (top-left/bottom-right) that were never present in the initial dataset. This environment allows us to assess whether an algorithm can escape the pull of a suboptimal data distribution to find the globally optimal policy. More details can be found in the Appendix D.

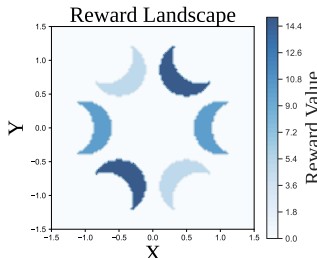

Figure 2: The visualization of the multi-crescent task.

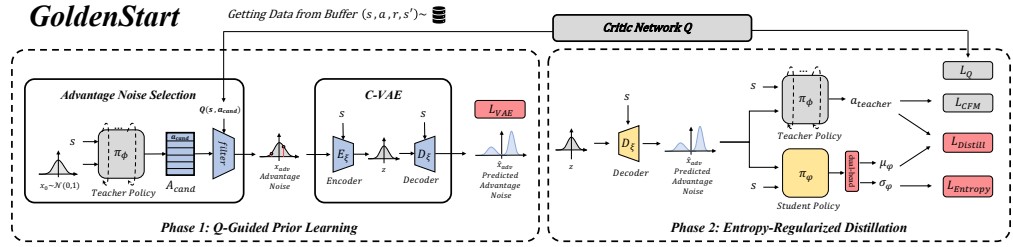

Figure 3: Overview of our algorithm. During training, we first learn a structured prior for the initial noise, which is then used to distill the teacher policy. For online exploration, actions are sampled from the student's entropy-regularized distribution. During evaluation, the deterministic mean of the policy's output is used. The critic update steps are omitted for clarity, detailed in Appendix B.

## 3 METHODOLOGY

### 3.1 OVERVIEW OF THE ALGORITHM

Our method, GS-flow, is designed to mitigate the two challenges of imprecise exploitation and ineffective exploration common in existing distilled policies through a two-phase training process, as illustrated in Figure 3. The first phase, Q-Guided Prior Learning, focuses on solving the suboptimal starting point problem. Instead of beginning the generation process from a standard, uninformed Gaussian noise, we use the Advantage Noise Selection module to actively identify advantage initial noises, which lead to high-value actions. We then train a conditional Variational Autoencoder (CVAE) to model the distribution of these advantage noises, effectively learning an informed, state-conditioned prior. The second phase, Entropy-Regularized Distillation, uses the learned prior to train a highly capable student policy. Both the teacher and student policies are provided with an initial noise sampled from our learned prior. Furthermore, the student model is designed as a stochastic policy and trained with a hybrid objective that combines distillation with an entropy regularization term. This endows the final actor with controllable stochasticity, allowing it to explore intelligently during online fine-tuning. The complete training pipeline, which integrates these two phases with standard actor-critic updates, is detailed in Algorithm 1.

At inference time, GS-flow operates with high efficiency using only the VAE decoder and the student policy, which are highlighted in yellow in Figure 3. Given the current state, the VAE decoder generates an advantage prior. This prior is then fed into the student actor to produce an action distribution. For online exploration, an action is sampled from this distribution with its learned mean and variance. For evaluation, we only use the mean to maximize exploitation.

---

**Algorithm 1** GS-flow

1: **Initialize:** Critic $Q_\theta$, VAE ($E_{\xi_1}, D_{\xi_2}$), Teacher Policy $\pi_\phi$, Student Policy $\pi_\varphi$.
2: **for** each training step **do**
3:     Sample a batch $\{(s, a, r, s')\}$ from dataset $\mathcal{D}$.
    # — 1. Update Critic —
4:     Update critic parameters $\theta$ using Temporal Difference (TD) learning.
    # — 2. Update Prior Learning Network —
5:     For each state $s$, generate $N_{\text{cand}}$ candidate actions $A_{\text{cand}} = \{a_j\}_{j=1}^{N_{\text{cand}}}$ using $\pi_\phi$.
6:     Find the prior noise $x_{\text{adv}}$ corresponding to the highest-Q action (Eq. 4).
7:     Update VAE parameters $\xi_1, \xi_2$ by minimizing the CVAE loss (Eq. 5).
    # — 3. Update Student Policy —
8:     Update teacher policy parameters $\phi$ using the flow matching loss (Eq. 1).
9:     Generate a sampled prior for the current state: $\hat{x}_{\text{adv}} \leftarrow D_{\xi_2}(s, \mathcal{N}(\mathbf{0}, \mathbf{I}))$.
10:     Generate the teacher's target action: $a_{\text{teacher}} \leftarrow \pi_\phi(s, \hat{x}_{\text{adv}})$.
11:     Update student policy parameters $\varphi$ by minimizing the actor loss $\mathcal{L}_{\text{Actor}}$ (Eq. 9).
12: **end for**
13: **return** Trained student policy $\pi_\varphi$.

---

## 3.2 Q-Guided Prior Learning

To realize our first insight of initiating the denoising process from golden starting points, we propose learning a Q-Guided Prior to model the distribution of what we named "advantage noises", denoted as $x_{\text{adv}}$. For this purpose, we employ a conditional Variational Autoencoder (CVAE) due to its flexibility in learning arbitrary multi-modal distributions. To achieve this, we first need to construct samples $\mathcal{B}_{\text{adv}}$ of these advantage noises for model training. A formal theoretical analysis is provided in Appendix C, where we analyze our method from the perspective of amortized optimization.

**Advantage Noise Selection.** We introduce a data collection module named Advantage Noise Selection, shown in Phase 1 of Figure 3. Given the state $s$, we first collect a set of $N_{\text{cand}}$ candidate actions, denoted as $a_{\text{cand}} \in A_{\text{cand}}$, generated by the teacher policy $\pi_\phi$ with $N_{\text{cand}}$ different initial noises $x_0$, which is sampled from a normal distribution:

$$A_{\text{cand}} = \{a_j = \pi_\phi(s, x_j) \mid x_j \sim \mathcal{N}(\mathbf{0}, \mathbf{I})\}_{j=1}^{N_{\text{cand}}}. \tag{3}$$

Although these candidate actions are all feasible behaviors learned from the dataset, they are not necessarily optimal. To identify the most promising starting point, we leverage the critic $Q$ to evaluate all candidate actions. The initial noise that generates the action with the highest Q-value is designated as the advantage noise for $s$:

$$x_{\text{adv}}(s) = \arg\max_{x_j} Q(s, \pi_\phi(s, x_j)). \tag{4}$$

This selection process is applied on-the-fly within each training step, using the most up-to-date teacher policy to generate a new batch of pairings, $\mathcal{B}_{\text{adv}} = \{(s, x_{\text{adv}}(s))\}$. This batch then serves as the target distribution for the CVAE update.

**State Conditional VAE.** With the data collected before, we then train a Conditional Variational Autoencoder (CVAE) (Kingma & Welling, 2013) to model the state-conditioned distribution $p_{\xi_2}(x_{\text{adv}}|s)$. The CVAE consists of a conditional encoder $E_{\xi_1}(x, s)$ and a conditional decoder $D_{\xi_2}(z, s)$, where $z$ is the latent vector. The encoder maps a prior-state pair to a latent distribution, while the decoder reconstructs the prior from a latent sample. The model is trained by minimizing the weighted sum of a reconstruction loss and a KL-divergence regularization term:

$$\mathcal{L}_{\text{VAE}}(\xi_1, \xi_2) = \mathcal{L}_{\text{recon}} + \lambda_{\text{KL}}\mathcal{L}_{\text{KL}}, \tag{5}$$

where $\lambda_{\text{KL}}$ is the scalar weight. The KL-divergence term $\mathcal{L}_{\text{KL}}$ regularizes the latent space by encouraging the encoded distribution to be close to a standard normal distribution $\mathcal{N}(0, I)$:

$$\mathcal{L}_{\text{KL}} = \mathbb{E}_{(s, x_{\text{adv}}) \sim \mathcal{B}_{\text{adv}}} \left[ D_{KL} \left( q_{\xi_1}(z \mid x_{\text{adv}}, s) \parallel \mathcal{N}(0, I) \right) \right], \tag{6}$$

where $q_{\xi_1}$ is the approximate posterior distribution, a diagonal Gaussian parameterized by the encoder $E_{\xi_1}$: $q_{\xi_1}(z \mid x_{\text{adv}}, s) = \mathcal{N}(\mu_{\xi_1}(x_{\text{adv}}, s), \Sigma_{\xi_1}(x_{\text{adv}}, s))$. Assuming $\hat{x}_{\text{adv}}$ denotes the prior predicted by $D_{\xi_2}(z, s)$, the loss of reconstruction $\mathcal{L}_{\text{recon}}$ can be calculated as follows:

$$\mathcal{L}_{\text{recon}} = \mathbb{E}_{(s, x_{\text{adv}}) \sim \mathcal{B}_{\text{adv}}, z \sim q_{\xi_1}(z|x_{\text{adv}}, s)} \left[ \|\hat{x}_{\text{adv}} - x_{\text{adv}}\|^2 \right]. \tag{7}$$

Notably, CVAE is capable of approximating an arbitrarily potentially multimodal prior distribution, offering an advantage over methods that learn a Gaussian distribution.

**Validation.** We validate the effectiveness of our Q-guided prior in the MultiCrescent environment. Figure 4 visualizes the distribution generated by the VAE decoder during inference. The red points represent $\hat{x}_{\text{adv}}$, and the red region generated via KDE (Silverman, 2018) represents the predicted prior distribution. After the offline phase (left panel), the prior captures the high-value modes (middle-left/middle-right) within the static dataset. After online fine-tuning (right panel), the prior adapts its density to focus on the newly discovered, globally optimal action modes (top-left/bottom-right). This demonstrates that our learned prior captures the distribution of advantage noises. Furthermore, Figure 5c shows that the actions generated from $\hat{x}_{\text{adv}}$ yield higher Q values compared to the baseline shown in Figure 5b.

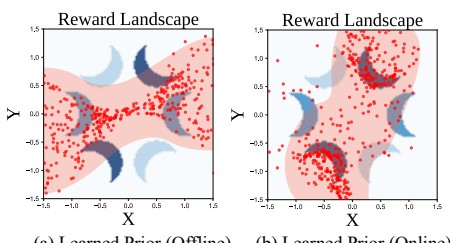

(a) Learned Prior (Offline)    (b) Learned Prior (Online)

Figure 4: Visualization of the learned prior distribution after different training stages.

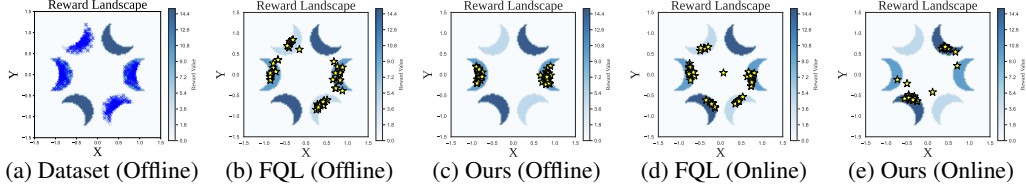

(a) Dataset (Offline)  (b) FQL (Offline)  (c) Ours (Offline)  (d) FQL (Online)  (e) Ours (Online)

Figure 5: Results on the multi-crescent task. Blue crosses denote samples from the offline dataset, while yellow stars represent the actions produced by the policies. **(a):** shows the offline dataset, which excludes the two globally optimal modes. **(b, c):** shows action distributions after the offline phase. Our method captures the higher-value modes within the dataset, while the baseline shows a less focused distribution. **(d, e):** shows action distributions after the online fine-tuning phase. Our method quickly discovers and converges to both highest-reward modes. In contrast, the baseline only finds one. More results on this task can be found in Appendix 8.

## 3.3 ENTROPY-REGULARIZED DISTILLATION

Previous flow-matching policy distillation methods produce a deterministic actor. Although efficient for exploitation, it is ill-suited for online exploration due to its lack of inherent stochasticity. This can be viewed as a point-to-point generation process, where a starting noise is mapped to a single target action. Inspired by recent approaches that augment generative models with distributional models (Dong et al., 2025), we propose an entropy-regularized distillation method. This transforms the distillation from a point-to-point mapping into a point-to-adaptive-distribution process, providing the agent with a principled method for balancing the exploration-exploitation trade-off.

To achieve this, we parameterize the student policy $\pi_\varphi(a|s, \hat{x}_{\text{adv}})$ as a Gaussian distribution using a dual-headed architecture that outputs both a mean $\mu_\varphi(s, \hat{x}_{\text{adv}})$ and a standard deviation $\sigma_\varphi(s, \hat{x}_{\text{adv}})$. The action $a_\varphi$ for exploration is computed as:

$$a_\varphi(s, \hat{x}_{\text{adv}}, \epsilon) = \mu_\varphi(s, \hat{x}_{\text{adv}}) + \sigma_\varphi(s, \hat{x}_{\text{adv}}) \odot \epsilon, \quad \text{where} \quad \epsilon \sim \mathcal{N}(0, I). \tag{8}$$

The actor policy is trained by minimizing a composite objective that balances three key components: imitation of the teacher, value maximization, and entropy regularization. The training objective for our entropy-regularized actor is a composite loss function designed to balance three key objectives: (1) imitating the high-quality teacher policy, (2) maximizing expected return according to the critic, and (3) maintaining sufficient policy entropy to encourage exploration. Therefore, with the advantage noise $\hat{x}_{\text{adv}} = D_{\xi_2}(z, s)$, the total actor loss is defined as follows:

$$\mathcal{L}_{\text{Actor}} = \mathbb{E}_{z \sim \mathcal{N}(0,I), s \sim \mathcal{D}} [\alpha_1 \mathcal{L}_{L2\text{-Distill}} + \mathcal{L}_Q - \alpha_2 \mathcal{H}(\pi_\varphi(\cdot|s, \hat{x}_{\text{adv}}))]. \tag{9}$$

**The distillation term** $\mathcal{L}_{L2\text{-Distill}}$ anchors the mean behavior of the student policy to the high-quality teacher actions, and $\alpha_1$ is the scalar weight to control BC behavior. Two details are critical to ensure that this process has a low-variance and stable training signal. First, both teacher and student policies are conditioned on identical advantage noise $\hat{x}_{\text{adv}}$. Second, the loss is computed using only the student's deterministic mean $\mu_\varphi(s, \hat{x}_{\text{adv}})$ rather than a stochastic sample. These design choices reduce the variance of the loss signal and improve training stability. The loss is defined as follows:

$$\mathcal{L}_{L2\text{-Distill}} = \mathbb{E}\left[\|\mu_\varphi(s, \hat{x}_{\text{adv}}) - \pi_\phi(s, \hat{x}_{\text{adv}})\|^2\right]. \tag{10}$$

**The value maximization term** $\mathcal{L}_Q$ encourages the policy to seek actions that the critic evaluates as having a high value (Fujimoto & Gu, 2021). Following the standard approach in Soft Actor-Critic (SAC) (Haarnoja et al., 2018), we use a sampled action $a_\varphi$ from the policy: $a_\varphi \sim \pi_\varphi(\cdot|s, \hat{x}_{\text{adv}})$. The loss is then calculated as its negative Q-value:

$$\mathcal{L}_Q = -Q(s, a_\varphi). \tag{11}$$

**The third entropy bonus term** $\mathcal{H}(\pi_\varphi(\cdot|s, \hat{x}_{\text{adv}}))$ is the entropy of the policy under $\hat{x}_{\text{adv}}$. To automate the trade-off between reward and entropy, the temperature parameter $\alpha_2$ is learned by minimizing a separate loss function that aims to match the entropy to a predefined target entropy $\mathcal{H}_{\text{target}}$. This allows the agent to dynamically adjust its stochasticity, exploring more when the entropy is below the target and exploiting more when it is sufficient. These two components are calculated as follows:

$$\mathcal{H}(\pi_\varphi(\cdot|s, \hat{x}_{\text{adv}})) = -\mathbb{E}_{a_\varphi \sim \pi_\varphi} [\log \pi_\varphi(a_\varphi|s, \hat{x}_{\text{adv}})], \tag{12}$$

$$\mathcal{L}_{\alpha_2} = \mathbb{E}_{s \sim \mathcal{D}} [\alpha_2(\mathcal{H}(\pi_\varphi(\cdot|s, \hat{x}_{\text{adv}}) - \mathcal{H}_{\text{target}}))]. \tag{13}$$

Table 1: Offline performance on OGBench and D4RL benchmarks, averaged over 5 seeds (3 for Visual Environments due to computational cost). Best results are in **bold**. The performance of baseline methods is reported from Park et al. (2025b).

| Task | Gaussian Policies | | | Diffusion Policies | | | Flow Policies | | | | |
|---|---|---|---|---|---|---|---|---|---|---|---|
| | BC | IQL | ReBRAC | IDQL | SRPO | CAC | FAWAC | FBRAC | IFQL | FQL | Ours |
| **_OGBench_** | | | | | | | | | | | |
| AntMaze Large Navigate | $0 \pm 0$ | $48 \pm 9$ | $\mathbf{91} \pm 10$ | $0 \pm 0$ | $0 \pm 0$ | $42 \pm 7$ | $1 \pm 1$ | $70 \pm 20$ | $24 \pm 17$ | $80 \pm 8$ | $88.4 \pm 2.7$ |
| AntMaze Giant Navigate | $0 \pm 0$ | $0 \pm 0$ | $\mathbf{27} \pm 22$ | $0 \pm 0$ | $0 \pm 0$ | $0 \pm 0$ | $0 \pm 0$ | $0 \pm 1$ | $0 \pm 0$ | $4 \pm 5$ | $10.4 \pm 5.9$ |
| HumanoidMaze Medium | $1 \pm 0$ | $32 \pm 7$ | $16 \pm 9$ | $1 \pm 1$ | $0 \pm 0$ | $38 \pm 19$ | $6 \pm 2$ | $25 \pm 8$ | $\mathbf{69} \pm 19$ | $19 \pm 12$ | $45.0 \pm 19.7$ |
| HumanoidMaze Large | $0 \pm 0$ | $3 \pm 1$ | $2 \pm 1$ | $0 \pm 0$ | $0 \pm 0$ | $1 \pm 1$ | $0 \pm 0$ | $0 \pm 1$ | $6 \pm 2$ | $\mathbf{7} \pm 6$ | $4.6 \pm 4.4$ |
| AntSoccer Arena | $1 \pm 0$ | $3 \pm 2$ | $0 \pm 0$ | $0 \pm 1$ | $0 \pm 0$ | $0 \pm 0$ | $12 \pm 3$ | $24 \pm 4$ | $16 \pm 9$ | $39 \pm 6$ | $\mathbf{46.0} \pm 10.5$ |
| Cube Single Play | $3 \pm 1$ | $85 \pm 8$ | $92 \pm 4$ | $96 \pm 2$ | $82 \pm 16$ | $80 \pm 30$ | $81 \pm 9$ | $83 \pm 13$ | $73 \pm 3$ | $\mathbf{97} \pm 2$ | $95.6 \pm 4.1$ |
| Cube Double Play | $0 \pm 0$ | $1 \pm 1$ | $7 \pm 3$ | $16 \pm 10$ | $0 \pm 0$ | $2 \pm 2$ | $2 \pm 1$ | $22 \pm 12$ | $9 \pm 5$ | $36 \pm 6$ | $\mathbf{51.3} \pm 6.2$ |
| Scene Play | $1 \pm 1$ | $12 \pm 3$ | $50 \pm 13$ | $33 \pm 14$ | $2 \pm 2$ | $50 \pm 40$ | $18 \pm 8$ | $46 \pm 10$ | $0 \pm 0$ | $76 \pm 9$ | $\mathbf{88.0} \pm 8.6$ |
| Puzzle-3x3 Play | $1 \pm 1$ | $2 \pm 1$ | $2 \pm 1$ | $0 \pm 0$ | $0 \pm 0$ | $0 \pm 0$ | $1 \pm 1$ | $2 \pm 2$ | $0 \pm 0$ | $16 \pm 5$ | $\mathbf{25.2} \pm 10.7$ |
| Puzzle-4x4 Play | $0 \pm 0$ | $5 \pm 2$ | $10 \pm 3$ | $\mathbf{26} \pm 6$ | $7 \pm 4$ | $1 \pm 1$ | $0 \pm 0$ | $5 \pm 1$ | $21 \pm 11$ | $11 \pm 3$ | $16.7 \pm 4.1$ |
| **Average** | 0.7 | 19.1 | 29.7 | 17.2 | 9.1 | 21.4 | 12.1 | 27.7 | 21.8 | 38.5 | **47.1** |
| **_D4RL AntMaze_** | | | | | | | | | | | |
| AntMaze U-Maze | 55 | 77 | 98 | 94 | 97 | $66 \pm 5$ | $90 \pm 6$ | $94 \pm 3$ | $92 \pm 6$ | $96 \pm 2$ | $\mathbf{99.6} \pm 0.8$ |
| AntMaze U-Maze Diverse | 47 | 54 | 84 | 80 | 82 | $66 \pm 11$ | $55 \pm 7$ | $82 \pm 9$ | $62 \pm 12$ | $89 \pm 5$ | $\mathbf{93.2} \pm 7.1$ |
| AntMaze Medium Play | 0 | 66 | $\mathbf{90}$ | 84 | 81 | $49 \pm 24$ | $52 \pm 12$ | $77 \pm 7$ | $56 \pm 15$ | $78 \pm 7$ | $77.2 \pm 9.0$ |
| AntMaze Medium Diverse | 1 | 74 | 84 | $\mathbf{85}$ | 75 | $0 \pm 1$ | $44 \pm 15$ | $77 \pm 6$ | $60 \pm 25$ | $71 \pm 13$ | $75.5 \pm 11.0$ |
| AntMaze Large Play | 0 | 42 | 52 | 64 | 54 | $0 \pm 0$ | $10 \pm 6$ | $32 \pm 21$ | $55 \pm 9$ | $84 \pm 7$ | $\mathbf{86.5} \pm 5.2$ |
| AntMaze Large Diverse | 0 | 30 | 64 | 68 | 54 | $0 \pm 0$ | $16 \pm 10$ | $20 \pm 17$ | $64 \pm 8$ | $83 \pm 4$ | $\mathbf{84.8} \pm 5.2$ |
| **Average** | 17.2 | 57.2 | 78.7 | 79.2 | 73.8 | 30.2 | 44.5 | 63.7 | 64.8 | 83.5 | **86.1** |
| **_Visual Environments_** | | | | | | | | | | | |
| Visual Cube Single Play | – | $70 \pm 12$ | $83 \pm 6$ | – | – | – | – | $55 \pm 8$ | $49 \pm 7$ | $81 \pm 12$ | $\mathbf{92.7} \pm 4.1$ |
| Visual Cube Double Play | – | $34 \pm 23$ | $4 \pm 4$ | – | – | – | – | $6 \pm 2$ | $8 \pm 6$ | $21 \pm 11$ | $\mathbf{42.0} \pm 11.8$ |
| Visual Scene Play | – | $97 \pm 2$ | $98 \pm 4$ | – | – | – | – | $46 \pm 4$ | $86 \pm 10$ | $98 \pm 3$ | $\mathbf{100.0} \pm 0.0$ |
| Visual Puzzle-3x3 | – | $7 \pm 15$ | $88 \pm 4$ | – | – | – | – | $7 \pm 2$ | $\mathbf{100} \pm 0$ | $94 \pm 1$ | $88.67 \pm 9.0$ |
| Visual Puzzle-4x4 | – | $0 \pm 0$ | $26 \pm 6$ | – | – | – | – | $0 \pm 0$ | $8 \pm 15$ | $\mathbf{33} \pm 6$ | $31.00 \pm 7.1$ |
| **Average** | – | 41.6 | 59.8 | – | – | – | – | 22.8 | 50.2 | 65.4 | **70.9** |

**Validation.** The online fine-tuning results, shown in Figures 5d and 5e, highlight the superiority of our exploration mechanism. Our method effectively explores the action space and identifies both of the highest Q-value peaks (top-left/bottom-right). In contrast, the baseline lacks an exploration strategy and finds just one of the peaks. Even better, our method finds both modes using significantly fewer samples. This demonstrates the clear efficiency of its entropy-regularized exploration.

## 4 EXPERIMENTS

### 4.1 EXPERIMENTAL SETUP

We test our method across the OGBench (Park et al., 2025a) tasks, the standard D4RL AntMaze (Fu et al., 2020) tasks, and a set of challenging Visual Environments (Park et al., 2025a). The baselines range from standard Gaussian policies (BC, IQL, ReBRAC) (Kostrikov et al., 2022; Tarasov et al., 2023), to more expressive Diffusion Policies (IDQL, SRPO, CAC) (Hansen-Estruch et al., 2023; Chen et al., 2024; Ding & Jin, 2024), and finally to Flow Policies (FAWAC, FBRAC, IFQL) (Nair et al., 2021; Wang et al., 2023; Park et al., 2025b). And state-of-the-art flow-matching distillation model FQL (Park et al., 2025b). For offline-to-online, we also compared with Cal-QL and RLPD (Nakamoto et al., 2023; Ball et al., 2023).

Notably, RLPD is an advanced reinforcement learning algorithm explicitly built upon and designed to improve the standard Soft Actor-Critic (SAC) for effectively utilizing offline data Ball et al. (2023); Haarnoja et al. (2018). While RLPD shares the same maximum entropy objective as SAC, it incorporates architectural enhancements to achieve more stable and efficient performance in offline-to-online transitions. To provide a more direct understanding of how our framework compares with standard continuous control methods, we provide a comparison between *Ours* and the original SAC in Appendix H.5.

We validate our model in both offline and offline-to-online fine-tuning settings to demonstrate the effectiveness of our two contributions. More details on environments and baselines are shown in Appendices E and F.

Table 2: Offline-to-online performance comparison. Similar to Table 1, we report the results over 5 seeds. The best online results are highlighted in **bold**.

| Task | IQL | ReBRAC | Cal-QL | RLPD | IFQL | FQL | Ours |
|---|---|---|---|---|---|---|---|
| HumanoidMaze Medium | $21 \pm 13 \to 16 \pm 8$ | $16 \pm 20 \to 1 \pm 1$ | $0 \pm 0 \to 0 \pm 0$ | $0 \pm 0 \to 8 \pm 10$ | $56 \pm 35 \to \mathbf{82} \pm 20$ | $12 \pm 7 \to 22 \pm 12$ | $45 \pm 20 \to 67 \pm 6$ |
| AntSoccer Arena | $2 \pm 1 \to 0 \pm 0$ | $0 \pm 0 \to 0 \pm 0$ | $0 \pm 0 \to 0 \pm 0$ | $0 \pm 0 \to 0 \pm 0$ | $26 \pm 15 \to 39 \pm 10$ | $28 \pm 8 \to \mathbf{86} \pm 5$ | $46 \pm 10 \to 77 \pm 9$ |
| Cube Double Play | $0 \pm 1 \to 0 \pm 0$ | $6 \pm 5 \to 28 \pm 28$ | $0 \pm 0 \to 0 \pm 0$ | $0 \pm 0 \to 0 \pm 0$ | $12 \pm 9 \to 40 \pm 5$ | $40 \pm 11 \to 92 \pm 3$ | $51 \pm 6 \to \mathbf{99} \pm 1$ |
| Scene Play | $14 \pm 11 \to 10 \pm 9$ | $55 \pm 10 \to \mathbf{100} \pm 0$ | $1 \pm 2 \to 50 \pm 53$ | $0 \pm 0 \to \mathbf{100} \pm 0$ | $0 \pm 1 \to 60 \pm 39$ | $82 \pm 11 \to \mathbf{100} \pm 1$ | $88 \pm 9 \to \mathbf{100} \pm 0$ |
| Puzzle-4x4 Play | $5 \pm 2 \to 1 \pm 1$ | $8 \pm 4 \to 14 \pm 35$ | $0 \pm 0 \to 0 \pm 0$ | $0 \pm 0 \to \mathbf{100} \pm 1$ | $23 \pm 6 \to 19 \pm 33$ | $8 \pm 3 \to 38 \pm 52$ | $17 \pm 4 \to \mathbf{100} \pm 0$ |
| **Average** | $8.4 \to 5.4$ | $17.0 \to 28.6$ | $0.2 \to 10.0$ | $0.0 \to 41.6$ | $23.4 \to 48.0$ | $34.0 \to 67.6$ | $49.4 \to \mathbf{88.6}$ |
| AntMaze U-Maze | $77 \to 96$ | $98 \to 75$ | $77 \to \mathbf{100}$ | $0 \pm 0 \to 98 \pm 3$ | $94 \pm 5 \to 96 \pm 2$ | $97 \pm 2 \to 99 \pm 1$ | $100 \pm 1 \to \mathbf{100} \pm 1$ |
| AntMaze U-Maze Diverse | $60 \to 64$ | $74 \to 98$ | $32 \to 98$ | $0 \pm 0 \to 94 \pm 5$ | $69 \pm 20 \to 93 \pm 5$ | $79 \pm 16 \to \mathbf{100} \pm 1$ | $93 \pm 7 \to 98 \pm 3$ |
| AntMaze Medium Play | $72 \to 90$ | $88 \to 98$ | $72 \to \mathbf{99}$ | $0 \pm 0 \to 98 \pm 2$ | $52 \pm 19 \to 93 \pm 2$ | $77 \pm 7 \to 97 \pm 2$ | $77 \pm 9 \to 98 \pm 1$ |
| AntMaze Medium Diverse | $64 \to 92$ | $85 \to \mathbf{99}$ | $62 \to 98$ | $0 \pm 0 \to 97 \pm 2$ | $44 \pm 26 \to 89 \pm 4$ | $55 \pm 19 \to 97 \pm 3$ | $76 \pm 11 \to 98 \pm 2$ |
| AntMaze Large Play | $38 \to 64$ | $68 \to 32$ | $32 \to \mathbf{97}$ | $0 \pm 0 \to 93 \pm 5$ | $64 \pm 14 \to 80 \pm 5$ | $66 \pm 40 \to 84 \pm 30$ | $86 \pm 5 \to 91 \pm 10$ |
| AntMaze Large Diverse | $27 \to 64$ | $67 \to 72$ | $44 \to 92$ | $0 \pm 0 \to 94 \pm 3$ | $69 \pm 6 \to 86 \pm 5$ | $75 \pm 24 \to 94 \pm 3$ | $85 \pm 5 \to \mathbf{96} \pm 4$ |
| **Average** | $56.3 \to 78.3$ | $80.0 \to 79.0$ | $53.2 \to \mathbf{97.3}$ | $0.0 \to 95.7$ | $65.3 \to 89.5$ | $74.8 \to 95.2$ | $86.2 \to 96.8$ |
| Pen Cloned | $84 \to 102$ | $74 \to 138$ | $-3 \to -3$ | $3 \pm 2 \to 120 \pm 10$ | $77 \pm 7 \to 107 \pm 10$ | $53 \pm 14 \to \mathbf{149} \pm 6$ | $71 \pm 6 \to 146 \pm 6$ |
| Door Cloned | $1 \to 20$ | $0 \to 102$ | $-0 \to -0$ | $0 \pm 0 \to 102 \pm 7$ | $3 \pm 2 \to 50 \pm 15$ | $0 \pm 0 \to 102 \pm 5$ | $1 \pm 1 \to \mathbf{105} \pm 4$ |
| Hammer Cloned | $1 \to 57$ | $7 \to 125$ | $0 \to 0$ | $0 \pm 0 \to 128 \pm 29$ | $4 \pm 2 \to 60 \pm 14$ | $0 \pm 0 \to 127 \pm 17$ | $10 \pm 3 \to \mathbf{132} \pm 5$ |
| Relocate Cloned | $0 \to 0$ | $1 \to 7$ | $-0 \to -0$ | $0 \pm 0 \to 2 \pm 2$ | $-0 \pm 0 \to 5 \pm 3$ | $0 \pm 1 \to 62 \pm 8$ | $0 \pm 0 \to \mathbf{63} \pm 12$ |
| **Average** | $21.5 \to 44.8$ | $20.5 \to 93.0$ | $-0.8 \to -0.8$ | $0.8 \to 88.0$ | $21.0 \to 55.5$ | $13.2 \to 110.0$ | $20.5 \to \mathbf{111.5}$ |

## 4.2 RESULTS AND ANALYSIS

**The Impact of the Learned Prior on Offline Performance.** The results in Table 1 demonstrate the effectiveness of our proposed method, GS-flow. GS-flow achieves new state-of-the-art performance on average, outperforming all baselines. This advantage is particularly pronounced on several tasks with multi-modal action spaces, where our method shows significant gains over the strong FQL baseline. According to the results in OGBench, the strength can be illustrated by the contrasting results on the Cube tasks. On Cube Single Play, a task with a relatively unimodal optimal policy, GS-flow performs comparably to the strong FQL baseline. In contrast, on the more complex Cube Double Play, which requires coordinating two objects and therefore presents a significantly more multimodal Q landscape, the offline score of GS-flow (51.3%) dramatically outperforms all competing methods. The contrast underscores our algorithm's specialized capability in multi-modal challenges. The advantage is further substantiated in other complex manipulation tasks such as Puzzle-3x3 and Puzzle-4x4, and in challenging locomotion environments such as HumanoidMaze Medium Navigate, where GS-flow achieves more than double the score of FQL. Furthermore, GS-flow achieves the highest average scores on the remaining two benchmarks, D4RL AntMaze and Visual Environments, demonstrating the effectiveness of our algorithm in the offline setting.

**Effective Online Exploration via Controllable Entropy.** The second key advantage of GS-flow, its capacity for effective online exploration, is enabled by its controllable entropy mechanism based on the output of the distribution actor. The performance improvements on online finesetuning shown in Table 2 are comparable, especially in tasks that require extensive exploration. The Puzzle-4x4 environment serves as a powerful case study. As noted by the authors of FQL (Park et al., 2025b), this task is particularly challenging for methods with limited exploration capabilities. The baseline FQL reflects this, improving from 8% to 38% after online training. In the contrast, GS-flow leverages its entropy-regularized stochastic policy to achieve a score from 17% to 100%, matching the performance of specialized online methods like RLPD (Ball et al., 2023). Furthermore, our method significantly outperforms RLPD on other complex tasks such as AntSoccer and Cube Double. These results demonstrate that our entropy-regularized distillation successfully combines the high performance of the teacher model with the advantage of principled, controllable exploration found in traditional Gaussian policies (Haarnoja et al., 2018). We believe the idea of moving beyond a "point-to-point" mapping to a "point-to-distribution" process is simple yet valuable, allowing GS-flow to effectively balance exploitation and exploration.

## 4.3 FURTHER ANALYSIS

**The Importance of the Learned Prior.** To analyze the impact of our proposed prior learning mechanism, we evaluate its performance while varying the number of candidate actions, $N_{cand}$, used in the Advantage Noise Selection module. As depicted in Figure 6a, there is a clear trend: increasing the number of candidates improves both sample efficiency and final performance. The red curve ($N_{cand} = 15$) achieves the highest return, while the green curve ($N_{cand} = 5$) learns more slowly. However, even with only five candidates, our method significantly outperforms the FQL (the

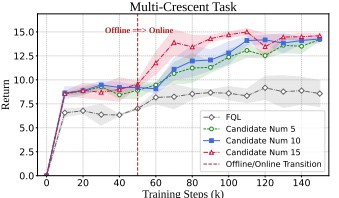 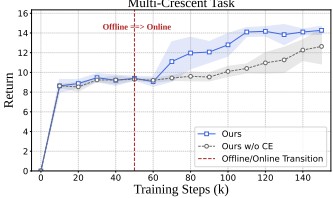 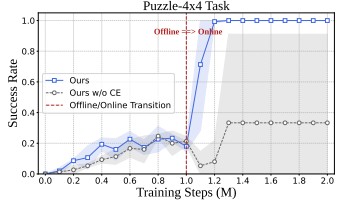

(a) Effect of the candidate number.     (b) Ablation on Multi-Crescent.     (c) Ablation on Puzzle-4x4.

Figure 6: Ablation studies on the offline-to-online transition. **(a):** The plot analyzes the impact of the candidate number on learning efficiency. **(b, c):** The plots demonstrate the effectiveness of our controllable entropy, showing significant performance gains of our full method over a deterministic variant in both the Multi-Crescent environment and the Puzzle-4x4 task.

gray curve), which can be viewed as a degenerate case of our approach without the Q-guided prior learning module($N_{cand} = 0$). This strongly validates the effectiveness of learning a structured prior. Given the trade-off between performance and computational overhead of the selection module, we chose $N_{cand} = 10$ (the blue curve) as a balanced setting for all main experiments.

**The Importance of the Controllable Entropy.** To isolate the contribution of our controllable entropy, we conduct ablation studies in the Multi-Crescent and Puzzle-4x4 environments. As shown in Figures 6b and 6c, our full method (the blue curve), which uses a dual-headed architecture with an entropy-regularized loss, demonstrates significantly higher learning efficiency during the online phase compared to a deterministic variant that uses only our learned prior module (the gray curve, denoted "Ours w/o CE"). In particular, the online performance of the gray curve in the Puzzle-4x4 task is similar to that of the FQL (as seen in Table 1). This similarity in performance strongly suggests that our controllable entropy mechanism is the key component that provides superior online exploration, effectively addressing this known limitation in prior work.

**Computational Cost Analysis.** We demonstrate that significant performance gains of our method do not come at a prohibitive computational cost, particularly during the critical inference phase. We compare the wall-clock time for a single training step and a single inference step against FQL and IFQL. As presented in Figure 7, the inference time of GS-flow(0.51 ms) is only marginally higher than that of FQL (0.42 ms), which is caused by the VAE Decoder model ($D_{\xi_2}$) and remains significantly faster than the multi-step IFQL (0.97 ms). This confirms that our method preserves the single-step efficiency of the

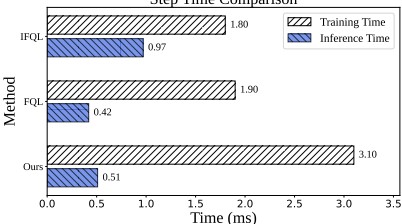

Figure 7: Average step time required on cube-double task.

distillation paradigm. Although the training time for GS-flow(3.10 ms) is higher due to the additional inference in the Advantage Noise Selection module under candidate number $N_{cand} = 10$, this one-time training cost is a well-justified trade-off for the substantial improvements in both policy quality and online adaptability.

## 5 RELATED WORKS

**Efficient Inference for Generative Policies.** Generative policies, including diffusion (Ho et al., 2020) and flow-matching models (Liu et al.; Albergo & Vanden-Eijnden, 2023; Geng et al., 2025), excel in representing multimodal action distributions in RL (Chi et al., 2023; Hansen-Estruch et al., 2023; Ding & Jin, 2024; Nair et al., 2021). However, their practical adoption is hindered by high inference latency (Shi & Zhang; Zhai et al., 2024). While one-step distillation methods (Park et al., 2025b) have improved inference speed, they often overlook the impact of the noise prior on optimization, a factor shown to be promising in the image generation field (Zhou et al., 2025). DSRL (Wagenmaker et al., 2025) takes advantage of this idea by learning a Gaussian prior distribution for online adaptation, without optimizing for inference latency. In contrast, our method integrates a more flexible prior while introducing negligible inference overhead.

**Online Exploration for Generative Policies.** Another key challenge for generative policies is principled online exploration (Fan et al., 2025). One line of research focuses on introducing stochasticity into the inference denoising process (Yang et al., 2023; Black et al., 2024b; Chen et al., 2025). Another line focuses on the training phase, using techniques such as reweighted score matching (Ma et al., 2025a) and entropy estimation with Gaussian Mixture Models, which can be computationally expensive (Wang et al., 2024). Recently, EXPO (Dong et al., 2025) enhances sample efficiency by training an additional Gaussian edit policy with entropy regularization. In contrast to these methods, our approach is more lightweight, integrating entropy control directly into the distillation process.

## 6 CONCLUSION

In this work, we introduced GS-flow, a novel framework for distilling flow-matching policies. Our method makes two key contributions: it learns a Q-Guided Generative Prior to provide a "golden start" that shortcuts the policy to high-value actions, and it uses Entropy-Regularized Distillation to endow the policy with controllable, principled exploration. Extensive experiments show that GS-flow establishes a new state-of-the-art in overall performance on challenging benchmarks, particularly excelling on complex tasks that require multi-modal actions and effective exploration. Our framework successfully bridges the gap between expressive generative models and practical actor-critic methods, delivering a potent combination of inference speed, precision, and exploratory power.

### ACKNOWLEDGMENTS

This work was supported in part by the National Key R&D Program of China (Grant No.2023YFF0725001), in part by the National Natural Science Foundation of China (Grant No.92370204, 62306255), in part by the guangdong Basic and Applied Basic Research Foundation (Grant No.2023B1515120057), in part by the Key-Area Special Project of Guangdong Provincial Ordinary Universities(2024ZDZX1007).

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

## A    USE OF LARGE LANGUAGE MODELS

We utilized Large Language Models as a tool to assist in the preparation of this paper. Specifically, LLMs were used for polishing the language, correcting grammar, and providing suggestions for LaTeX formatting to improve the manuscript's presentation. Our use of LLMs is in full compliance with the ICLR 2026 policy. We reviewed all LLM-generated outputs and take full responsibility for the scientific claims and all content in this work.

## B    CRITIC UPDATE DETAILS

Our critic network, denoted as $Q_\theta(s, a)$, is trained to estimate the expected discounted cumulative reward (the Q-value) for taking action $a$ in state $s$. To improve training stability and mitigate the overestimation of Q-values, we employ standard techniques from modern actor-critic methods (Fujimoto et al., 2018; Haarnoja et al., 2018). Specifically, we use a twin-critic architecture, maintaining two separate Q-networks ($Q_{\theta_1}, Q_{\theta_2}$), and use slowly-updated target networks ($Q_{\theta'_1}, Q_{\theta'_2}$) to construct the Bellman target. The critic parameters are optimized by minimizing the Mean Squared Bellman Error (MSBE). For a given transition $(s, a, r, s')$ from the replay buffer, we first compute the target value, $y$. The next action, $a'$, is sampled from our stochastic student policy, $\pi_\varphi$, and the target value includes an entropy term to maintain consistency with the actor's objective:

$$y = r + \gamma \left( \min_{i=1,2} Q_{\theta'_i}(s', a') - \alpha_2 \log \pi_\varphi(a'|s') \right), \quad \text{where} \quad a' \sim \pi_\varphi(\cdot|s'). \tag{14}$$

The total loss for the critic networks is the sum of the MSBE for each critic with respect to this common target value:

$$\mathcal{L}_{\text{Critic}}(\theta_1, \theta_2) = \mathbb{E}_{(s,a,r,s') \sim \mathcal{D}} \left[ \sum_{i=1,2} (Q_{\theta_i}(s, a) - y)^2 \right]. \tag{15}$$

The target network parameters $\theta'$ are updated via Polyak averaging with the main critic parameters $\theta$ at each training step: $\theta' \leftarrow \tau\theta + (1 - \tau)\theta'$, where $\tau$ is a small interpolation factor.

## C    THEORETICAL ANALYSIS: AMORTIZED OPTIMIZATION BOUND

In this section, we provide a formal theoretical analysis for the performance improvement of our method. We frame our approach as an instance of **Amortized Optimization**, where the computationally expensive test-time selection procedure—searching for the optimal noise vector—is "amortized" into a learned generative model (CVAE).

### C.1    ORACLE POLICY VIA REJECTION SAMPLING

Consider an idealized oracle policy that performs Best-of-$N$ sampling at test time. For a given state $s$, it draws $N$ noise vectors $\{x_i\}_{i=1}^{N} \sim \mathcal{N}(0, I)$ and selects the one that maximizes the predicted value:

$$x_N^* = \arg\max_{x_i} Q(s, \pi(s, x_i)). \tag{16}$$

By standard order-statistics results, the expected return of this oracle is strictly superior to a single random sample (the baseline policy):

$$J_{\text{oracle}} = \mathbb{E}[Q(s, \pi(s, x_N^*))] \geq \mathbb{E}[Q(s, \pi(s, x))] = J_{\text{base}}. \tag{17}$$

We define this non-negative gain as the *Selection Gain*: $\Delta_{\text{select}} = J_{\text{oracle}} - J_{\text{base}} \geq 0$.

### C.2    AMORTIZING THE ORACLE VIA CVAE

To bypass the prohibitive computational cost of running the oracle during online inference, we train a CVAE to approximate the distribution of the oracle-selected noise vectors $x_N^*$. The training objective minimizes:

$$\mathcal{L}_{\text{VAE}} = \mathcal{L}_{\text{recon}} + \lambda \mathcal{L}_{\text{KL}}, \tag{18}$$

where the reconstruction error is defined as $\epsilon_{\text{vae}} = \mathbb{E}\|x_N^* - \hat{x}\|^2$, with $\hat{x} \sim p_\xi(x|s)$ sampled from the learned CVAE prior.

## C.3 Performance Bound

Let $V(x) = Q(s, \pi(s, x))$ be the value associated with a noise vector $x$. The expected return of our amortized policy is $J_{\text{amortized}} = \mathbb{E}_{\hat{x} \sim p_\xi}[V(\hat{x})]$. We assume $V(x)$ is $L$-Lipschitz continuous with respect to the noise vector $x$, i.e., $|V(x) - V(y)| \leq L\|x - y\|$. The performance gap between the oracle and our policy can be bounded as:

$$J_{\text{oracle}} - J_{\text{amortized}} = \mathbb{E}[V(x_N^*) - V(\hat{x})] \leq L \cdot \mathbb{E}\|x_N^* - \hat{x}\|. \tag{19}$$

Applying Jensen's inequality ($\mathbb{E}[X] \leq \sqrt{\mathbb{E}[X^2]}$), we obtain:

$$J_{\text{oracle}} - J_{\text{amortized}} \leq L\sqrt{\mathbb{E}\|x_N^* - \hat{x}\|^2} = L\sqrt{\epsilon_{\text{vae}}}. \tag{20}$$

Substituting $J_{\text{oracle}} = J_{\text{base}} + \Delta_{\text{select}}$, we derive the final lower bound for our method:

$$J_{\text{amortized}} \geq \underbrace{J_{\text{base}}}_{\text{Baseline}} + \underbrace{\Delta_{\text{select}}}_{\text{Selection Gain}} - \underbrace{L\sqrt{\epsilon_{\text{vae}}}}_{\text{Amortization Gap}}. \tag{21}$$

## C.4 Interpretation and Empirical Alignment

The derived bound offers a theoretical explanation for the empirical phenomena observed in our experiments:

- **Effect of $N$ ($\Delta_{\text{select}}$):** As the number of candidates $N$ increases, the expected maximum of the samples increases, raising $\Delta_{\text{select}}$. This aligns with our findings in Figure 6(a), where performance improves consistently with larger $N$ used during prior training.

- **Robustness to $Q$-landscape ($L$):** The Lipschitz constant $L$ reflects the smoothness of the value landscape. In complex environments like the Multi-Crescent toy task, the selection gain $\Delta_{\text{select}}$ remains sufficient to maintain performance gains despite the challenges of non-convex landscapes.

- **Importance of Reconstruction ($\epsilon_{\text{vae}}$):** The bound explicitly shows that minimizing the CVAE reconstruction error maximizes the return's lower bound. This is corroborated by results in Table 3, where lower VAE loss directly translates to higher returns.

## D Additional Studies in the Multi-Crescent Environment

To further validate the effectiveness of our custom Multi-Crescent Environment at highlighting key algorithmic challenges, we conducted a broader set of experiments with different baseline settings, as shown in Figure 8 in the main text. In this analysis, the gray bars represent the final offline training performance, while the blue bars show the performance after the subsequent online fine-tuning phase. The hyperparameter $\alpha$ corresponds to the weight of the behavioral cloning (BC) term in the FQL loss function (Equation 2). A smaller $\alpha$ places a relatively larger emphasis on the Q-maximization term. The primary results in the main body compare our method against FQL with a high BC weight ($\alpha = 100$).

By lowering $\alpha$, we test the hypothesis that our non-convex environment can induce Q-value overestimation in the baseline. The experimental results confirm this hypothesis. When $\alpha = 1$, FQL's offline performance drops sharply. The policy learns to target points overestimated by the critic—locations between the tips of the crescent shapes but outside the actual high-reward regions—leading to a significant decrease in return. In the extreme case where $\alpha = 0$ (i.e., pure Q-maximization), the offline return predictably falls to zero, further demonstrating our environment's ability to challenge methods susceptible to Q-value overestimation. The final two columns in the figure are designed to evaluate the performance of pure imitation learning. The "FQL w/o Q" baseline isolates the effect of imitation learning via flow-matching, while "BC (MSE)" represents a standard behavioral cloning approach. Our method outperforms both of these baselines. Notably, the standard FQL provides only a marginal improvement over "FQL w/o Q." This is because the reward modes in our environment are disconnected; when the Q-guidance is too weak (high $\alpha$), the policy struggles to jump from one mode to another, and when it is too strong (low $\alpha$), the policy is misled by critic overestimation. In contrast, our algorithm learns an initial noise distribution that directly fits the inherently high-Q actions from the dataset, making it significantly more robust to the effects of Q-value overestimation.

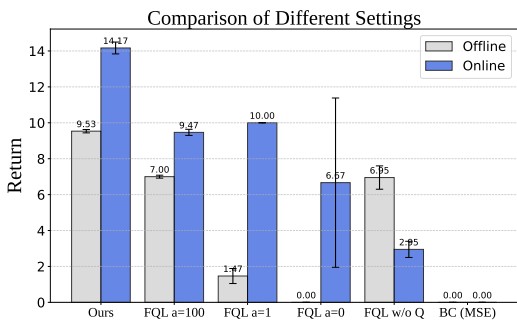

Figure 8: Additional Studies in the Multi-Crescent Environment

# E    BENCHMARK DESCRIPTIONS

## E.1    OGBENCH

OGBench (Offline Goal-Conditioned RL Benchmark) Park et al. (2025a) is a high-quality benchmark designed for offline goal-conditioned reinforcement learning. It aims to systematically evaluate the capabilities of algorithms across several key dimensions, such as trajectory stitching, long-horizon reasoning, handling high-dimensional inputs (e.g., pixels), and coping with environmental stochasticity. We utilize a variety of environments from OGBench in our experiments, spanning locomotion, manipulation, and visual tasks. Notably, we evaluate on the default task for each environment. For instance, in the `cube-double-play` environment, we exclusively use the `cube-double-play-singletask-task2-v0` task, which can be found in Park et al. (2025a;b).

The specific environments used in our work include:

- **AntMaze and HumanoidMaze**: These are maze navigation tasks requiring an agent to control a complex quadruped robot (Ant) or a 21-DoF humanoid robot (Humanoid), respectively, to reach a target location. We employ various maze layouts, including "Large" and "Giant", with the "Navigate" dataset type to test long-horizon planning and control capabilities.

- **AntSoccer**: This is a more challenging locomotion task that requires the Ant agent to dribble a soccer ball while navigating. We use the "Arena" (open-field) version of this environment.

- **Cube**: This is a robotic manipulation task involving multi-block pick-and-place operations. The agent must move, stack, or swap single or multiple cubes according to a goal configuration. We use the "Single Play" and "Double Play" versions to test the agent's ability to learn generalizable multi-object manipulation skills from unstructured, random trajectories.

- **Scene**: This is a complex sequential manipulation task requiring the robot arm to interact with various household objects, including a drawer, a window, button locks, and a cube. It is designed to challenge the agent's sequential and long-horizon reasoning abilities.

- **Puzzle**: In this task, a robot arm must solve a "Lights Out" puzzle. The agent presses buttons on a grid to toggle the color of the pressed button and its neighbors to match a goal configuration. We use the 3×3 and 4×4 grid versions to specifically test for combinatorial generalization.

- **Visual Environments**: Many tasks in OGBench, particularly the manipulation suite, support both state-based and pixel-based inputs. We evaluate our methods in the corresponding visual environments (e.g., Visual Cube, Visual Scene, Visual Puzzle), which require the agent to learn control policies directly from 64×64 RGB images.

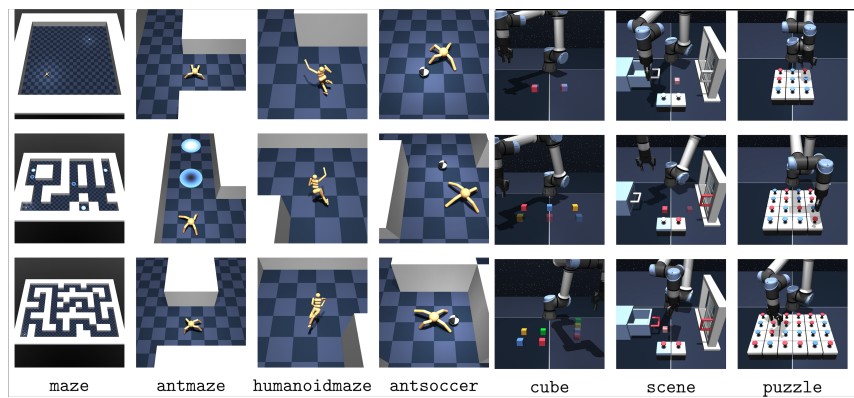

Figure 9: Visualization of the OGBench tasks.

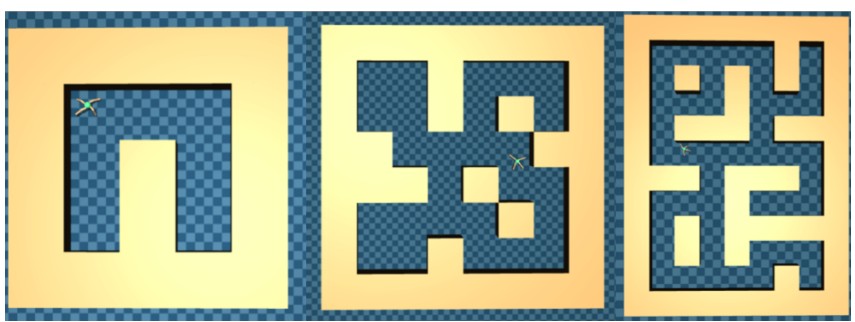

Figure 10: Visualization of the D4RL tasks.

## E.2 D4RL

D4RL (Datasets for Deep Data-Driven Reinforcement Learning) Fu et al. (2020) is a public benchmark focused on offline reinforcement learning. It is designed to provide datasets that reflect challenges present in real-world applications, such as narrow data distributions, undirected multi-task data, sparse rewards, and suboptimal data. These characteristics make D4RL an essential benchmark for evaluating the robustness and generalization of offline RL algorithms.

Our experiments primarily make use of the **AntMaze** environment from D4RL. This is a popular navigation task that requires an 8-DoF 'Ant' quadruped robot to reach a specified goal in a maze. The task features sparse rewards (a reward is only given upon reaching the goal), and the datasets are generated by a non-Markovian controller. This setup is designed to test an algorithm's ability to stitch effective trajectories from undirected data to solve long-horizon, sparse-reward tasks. We also implement on the Adroit domain, which involves controlling a 24-DoF robotic hand. Task examples are shown in Figure 10.

## F DETAILS ON ADDITIONAL BASELINE METHODS

In this section, we provide additional details on the baseline methods used in the paper (except FQL, which is introduced in Preliminary) , categorized by their underlying policy structure and learning paradigm. The settings for all baseline methods are adopted directly from the original paper (Park et al., 2025b) for comparison. You can find more implement details in their paper.

### F.1 OFFLINE RL BASELINES

For the offline RL experiments, we compare with 10 recent and representative methods to demonstrate our contributions.

**Gaussian Policies.** For standard offline RL methods that use Gaussian policies, we consider **BC**, **IQL** (Kostrikov et al., 2022), and **ReBRAC** (Tarasov et al., 2023). In particular, ReBRAC is known to perform well on many D4RL tasks (Fu et al., 2020), which are based on a behavior-regularized actor-critic framework.

**Diffusion Policies.** For methods based on diffusion policies, we compare against **IDQL** (Hansen-Estruch et al., 2023), **SRPO** (Chen et al., 2024), and Consistency-AC (**CAC**) (Ding & Jin, 2024). These methods employ different policy extraction techniques: IDQL is based on rejection sampling, whereas SRPO and CAC utilize policy distillation. CAC trains the distillation policy within the behavior-regularized actor-critic framework and is based on consistency models.

**Flow Policies.** We also consider several flow-based variants of existing algorithms to cover different policy extraction schemes. Flow Advantage-Weighted Actor-Critic (**FAWAC**) is a flow-based variant of AWAC (Nair et al., 2021), which uses the Advantage-Weighted Regression (AWR) objective for policy learning. Flow Behavior-Regularized Actor-Critic (**FBRAC**) is the flow counterpart to Diffusion-QL (DQL) (Wang et al., 2023), which is based on the original Q-loss with backpropagation through time. Implicit Flow Q-Learning (**IFQL**) is the flow counterpart to IDQL, based on a rejection sampling scheme.

## F.2 OFFLINE-TO-ONLINE FINETUNING BASELINES

The offline methods include **IQL**, which learns a policy implicitly through advantage-weighted regression over learned Q and Value functions; **ReBRAC**, a stable behavior-regularized actor-critic algorithm; and **IFQL**, a flow-based policy utilizing rejection sampling. We also include two methods designed for data-driven online RL: **Cal-QL** (Nakamoto et al., 2023), which calibrates the Q-function with the offline dataset to enable safer online exploration, and **RLPD** (Ball et al., 2023), which employs a balanced sampling strategy from both offline and online data buffers to accelerate fine-tuning.

## G HYPERPARAMETERS

### G.1 HYPERPARAMETERS SETTINGS

Table 4 lists the hyperparameters used for the cube-double experiment, based on the provided execution command.

The Offline Alpha and Online Alpha refer to the pre-set values of the hyperparameter $\alpha_1$ in Equation 9 for the offline-to-online transition. This distinction is made because the confidence in the critic's estimates differs between the offline and online phases. It is a common phenomenon in offline RL that the critic often overestimates Q-values, necessitating regulation of the Behavior Cloning (BC) weight. However, during the online phase, excessive reliance on the BC term can stifle exploration, which is why these values are set in advance.

Additionally, when running the online phase for the puzzle environment, we utilized the balanced sampling technique from Ball et al. (2023). To ensure a fair comparison, we also applied this technique to our FQL agent. We found that only our method showed performance improvements with this technique.

### G.2 THE EFFECT OF LATENT DIMENSION.

We investigate the sensitivity of our learned prior to the VAE's latent dimension. Figure 11 illustrates the results. We observe that a low-dimensional, compact latent space is optimal for this task, with performance peaking at a dimension of 1 or 2 for both offline and online settings. As the latent dimension increases, performance gradually degrades, particularly during the online phase. This suggests that a higher-dimensional space may increase the difficulty of learning a meaningful prior, potentially introducing noise or leading to overfitting. However, our method still outperforms the FQL across different tested dimensions in both settings. This demonstrates the fundamental robustness and benefit of our learned prior, even when its key hyperparameter is not perfectly tuned.

Table 3: Hyperparameters for the cube-double experiment.

| Hyperparameter | Value |
|---|---|
| Offline Steps | 1,000,000 |
| Online Steps | 1,000,000 |
| Seed | 0,2,4,8,16 |
| Latent Dimension | 8 (default) |
| KL Weight | 0.1 (default) |
| Reconstruction Weight | 1 (default) |
| Number of Candidates | 10 (default) |
| Offline Alpha1 | 300 |
| Online Alpha1 | 50 |
| Offline Temperature | 0 (default) |
| Target Entropy Multiplier | 0.5 (default) |

Table 4: Hyperparameters for the Puzzle 3x3 experiment.

| Hyperparameter | Value |
|---|---|
| Offline Steps | 1,000,000 |
| Online Steps | 1,000,000 |
| Seed | 0,2,4,8,16 |
| Latent Dimension | 8 (default) |
| KL Weight | 0.1 (default) |
| Reconstruction Weight | 1 (default) |
| Number of Candidates | 10 (default) |
| Offline Alpha1 | 1000 |
| Online Alpha1 | 10 |
| Offline Temperature | 0 (default) |
| Target Entropy Multiplier | 0.5 (default) |
| Balanced Sampling( Ball et al. (2023)) | True |

## H ADDITIONAL EXPERIMENTS

### H.1 COMPARISON TO REJECTION SAMPLING (RS)

We demonstrate that Ours is better, not merely faster, than Rejection Sampling (RS). While RS relies on stochastic filtering at inference time, Ours actively optimizes the generative process during training. We highlight the superiority of our approach regarding stability, robustness against overestimation, and safer distillation.

**1. Superior Stability and Precision.** Ours achieves significantly lower variance compared to IFQL. As shown in Table 5, IFQL relies on probabilistic sampling from a standard Gaussian distribution, which makes it difficult to guarantee high-quality actions in every iteration, leading to higher variance. In contrast, Ours learns a structured Q-guided prior via a CVAE. As the CVAE loss minimizes, the prior fits the distribution of high-value actions, allowing for stable generation without redundant sampling.

Table 5: Comparison of stability and precision between IFQL and Ours (Reward Avg/Std).

| Method | IFQL ($N = 24$) | IFQL ($N = 28$) | IFQL ($N = 32$) | Ours |
|---|---|---|---|---|
| Reward (Avg) | 7.6 | 8.1 | 8.0 | **9.2** |
| Reward (Std) | 1.5 | 1.6 | 1.5 | **0.2** |

**2. Robustness to Q-Value Overestimation.** The data in Table 5 shows that for IFQL, increasing $N$ from 28 to 32 does not yield further improvements ($8.1 \to 8.0$). In RS, increasing $N$ increases

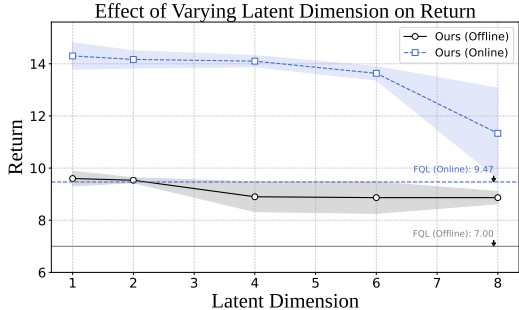

Figure 11: The effect of the VAE's latent dimension on the final return in both offline (black) and online (blue) settings.

the probability of selecting Out-of-Distribution (OOD) actions where the critic network erroneously predicts high Q-values (overestimation). In contrast, our Advantage Noise Selection is anchored to the data distribution. The target "advantage noises" are derived exclusively from the teacher policy, which is strictly constrained to the offline dataset's support.

**3. Further Analysis on Robustness.** To isolate the benefit of our distillation framework, we compared Ours against an FQL agent enhanced with Rejection Sampling ($N = 32$) on the *Cube-Double* task, where Q-overestimation is severe. As shown in Table 6, even with RS added, FQL fails to match Ours (37 vs. 51).

Table 6: Success Rate (%) on the Cube-Double task, comparing Ours with FQL and FQL-RS.

| Method | IFQL | FQL | FQL-RS ($N = 32$) | Ours |
|---|---|---|---|---|
| Success Rate (Avg) | 9 | 36 | 37 | **51** |
| Success Rate (Std) | **5** | 6 | 7 | 6 |

Ours is structurally superior from two perspectives:

- **Gradient Alignment:** In FQL, the distillation loss pulls the policy towards the teacher's *average* behavior, while the Q-loss pulls it towards the *mode* (highest value). These directions often conflict. Since RS merely filters outputs at inference time, it cannot mitigate this training conflict. In Ours, the distillation target is pre-selected for high value, ensuring the distillation and Q-maximization objectives are aligned.

- **Input Manifold Constraint:** FQL allows the optimizer to warp the mapping from any random noise to high-Q actions, risking overfitting to noise vectors that trigger critic overestimation. Ours constrains the student's input to the learned prior (CVAE), restricting the student to a "safe" latent manifold known to generate valid, in-distribution actions.

## H.2 SENSITIVITY ANALYSIS OF $\alpha_1$ AND REWARD HACKING

To further examine whether our Q-guided prior amplifies overestimation artifacts, we conducted a sensitivity analysis on the BC weight $\alpha_1$. We measured the bias between the predicted Q and true return $r$ (Bias $= Q - r$) on the Multi-Crescent environment.

The results in Table 7 confirm that reducing the BC constraint ($\alpha_1 \to 0$) leads to severe overestimation. However, with a reasonable setting ($\alpha_1 = 100$), Ours exhibits only marginal overestimation compared to the baseline (0.4 vs. 0.3) while achieving significant performance gains. This indicates that our Q-guided prior directs the agent to legitimate high-value regions rather than hacking the reward function.

Table 7: Analysis of Q-Overestimation and Robustness. Corresponding trends and performance comparisons are also illustrated in Figure 12.

| Model Setting | Ours ($\alpha_1 = 0$) | Ours ($\alpha_1 = 1$) | Ours ($\alpha_1 = 10$) | Ours ($\alpha_1 = 100$) | Baseline ($\alpha_1 = 100$) |
|---|---|---|---|---|---|
| Bias ($Q - r$) | 10.0 | 7.0 | 1.1 | 0.4 | 0.3 |
| Abs Bias ($|Q - r|$) | 10.0 | 7.0 | 1.1 | 0.4 | 0.3 |
| Return | - | - | - | **9.3** | 7.0 |

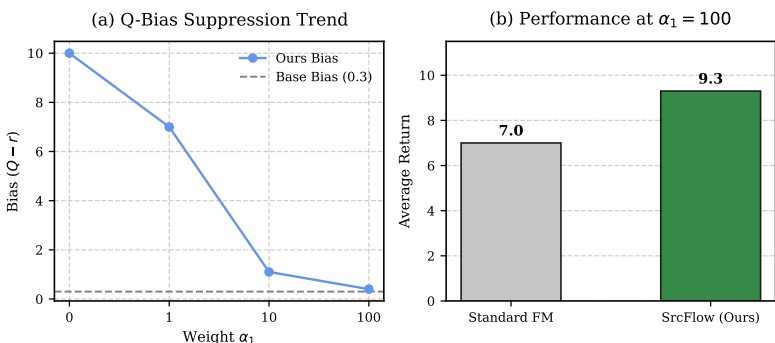

Figure 12: Q-bias suppression and performance gain.

## H.3 SYNERGY BETWEEN ENTROPY REGULARIZATION AND PRIOR LEARNING

To investigate the interaction between the entropy temperature $\alpha_2$ and the learned Q-guided prior during online fine-tuning, we conducted an ablation study in the Multi-Crescent environment. We varied the hyperparameter `target_entropy_multiplier` (denoted as $mult$), where the target entropy is defined as $H_{\text{target}} = -mult \times$ action_dim. A larger $mult$ results in a lower target entropy, thereby reducing the incentive for exploration.

We initialized all variants from the same offline checkpoint and performed online fine-tuning for 100k steps. The results, including Return, final $\alpha_2$, and VAE Loss, are summarized in Table 8.

Table 8: Interaction analysis between entropy and Q-guided prior on Multi-Crescent Environment. Note that variations in VAE Loss are primarily driven by the KL divergence component.

| Setting | Target Entropy | Return | Final $\alpha_2$ | VAE Loss |
|---|---|---|---|---|
| Ours (High Exp.) | High ($mult = 0.1$) | **14.0** | 3.6 | **0.090** |
| Ours (Med Exp.) | Medium ($mult = 0.5$) | 12.8 | 2.3 | 0.093 |
| Ours (Low Exp.) | Low ($mult = 0.75$) | 9.3 | 1.9 | 0.117 |
| Ours w/o Q-Prior | High ($mult = 0.1$) | 5.9 | 2.1 | - |

**Analysis of Mutual Benefit.** The experimental results yield three key insights regarding the synergistic relationship between these two components:

- **Q-Prior as the Foundation:** The variant without the Q-guided prior suffers a drastic performance drop (from 14.0 to 5.9). This confirms that the Q-prior provides a robust distribution that ensures a valid performance lower bound and protects the student policy from Out-of-Distribution (OOD) inputs during early fine-tuning.

- **Entropy Drives Local Optimization:** While the Q-prior successfully points the agent toward general high-value regions, the entropy term ($\alpha_2$) is essential for driving the local exploration required to find the optimal solution within those regions. Increasing the exploration incentive ($mult = 0.75 \rightarrow 0.1$) leads to a substantial increase in Return (9.3 to 14.0).

- **Entropy Assists Prior Fitting:** We observe a strong correlation where higher entropy leads to lower VAE loss. When exploration is constrained (Low Exploration), the policy tends to

collapse into sharp, potentially suboptimal local modes that are difficult for the VAE prior to model, resulting in a higher KL loss (0.117). Conversely, when exploration is sufficient, $\alpha_2$ smooths the policy distribution. This smoothing effect prevents mode collapse and makes high-value actions easier for the prior to fit, resulting in the lowest VAE loss (0.090).

In conclusion, entropy regularization actively assists the prior learning process by maintaining a learnable distribution landscape, proving that the two modules are synergistic rather than competing during online adaptation.

### H.4 QUANTIFICATION OF EXPLORATION CAPABILITY

**Quantitative Validation on Puzzle-4x4.** We further evaluate exploration efficiency on the `Puzzle-4x4` task. This environment consists of a $4 \times 4$ matrix where the objective is to turn off all lights (set the matrix values to 0) by toggling states. To measure the extent of state-space coverage during exploration, we define the `max_button_state` metric as the maximum value of the matrix mean observed during the entire online exploration process. A higher value indicates that the agent has reached states that are significantly different from the initial or goal states, reflecting a broader exploration scope. The results, averaged over 3 random seeds, are presented in Figure 13.

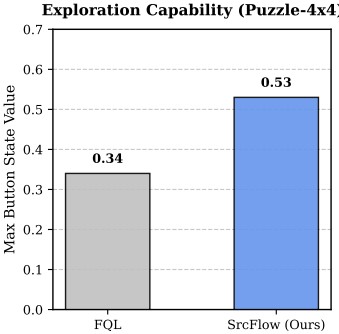

Figure 13: Exploration capability on the Puzzle-4x4 task. Ours reaches states with significantly higher `max_button_state`, indicating a wider exploration of the state space.

**Interpretation of Results.** As shown in Figure 13, Ours achieves a `max_button_state` of 0.53, which is significantly higher than FQL's 0.34. These maximum values primarily occur during the early stages of online interaction. This quantitative gap confirms that our algorithm indeed visits diverse and novel states that the baseline fails to reach.

### H.5 DIRECT COMPARISON WITH SOFT ACTOR-CRITIC (SAC)

To further evaluate the effectiveness of our entropy-regularized distillation framework, we conducted a direct comparison against the original Soft Actor-Critic (SAC) and its offline-to-online variant, RLPD, on OGBench and D4RL AntMaze tasks. SAC is a standard baseline for continuous control, while RLPD is a high-performance algorithm designed to improve SAC for utilizing offline data.

As shown in Table 9, SAC struggles significantly in most OGBench tasks, often failing to make progress (0% success rate). This is primarily due to the multi-modal action spaces and sparse rewards in these environments, where a standard Gaussian actor cannot effectively leverage the provided offline demonstrations.

**Analysis.** The results demonstrate that Ours achieves far superior performance on the most complex tasks:

- **Handling Multi-modality:** In tasks like *AntSoccer Arena* and *Cube Double Play*, both SAC and RLPD fail completely (0% success rate). This highlights a critical limitation of Gaussian actors in tasks requiring complex, multi-modal coordination. Ours, by distilling a

Table 9: Performance comparison (Success Rate %) during the Offline $\rightarrow$ Online transition. All results are averaged over 3 random seeds.

| Task | SAC (Off $\rightarrow$ On) | RLPD (Off $\rightarrow$ On) | Ours (Off $\rightarrow$ On) |
|---|---|---|---|
| HumanoidMaze Medium | $0 \rightarrow 2$ | $0 \rightarrow 84$ | $5 \rightarrow 67$ |
| AntSoccer Arena | $0 \rightarrow 0$ | $0 \rightarrow 0$ | $46 \rightarrow 77$ |
| Cube Double Play | $0 \rightarrow 0$ | $0 \rightarrow 0$ | $51 \rightarrow 99$ |
| Scene Play | $0 \rightarrow 100$ | $0 \rightarrow 100$ | $88 \rightarrow 100$ |
| Puzzle-4x4 Play | $0 \rightarrow 0$ | $0 \rightarrow 100$ | $17 \rightarrow 100$ |
| AntMaze U-Maze | $0 \rightarrow 100$ | $0 \rightarrow 98$ | $100 \rightarrow 100$ |
| AntMaze U-Maze Diverse | $0 \rightarrow 100$ | $0 \rightarrow 94$ | $93 \rightarrow 98$ |
| AntMaze Medium Play | $0 \rightarrow 98$ | $0 \rightarrow 98$ | $77 \rightarrow 98$ |
| AntMaze Medium Diverse | $0 \rightarrow 98$ | $0 \rightarrow 97$ | $76 \rightarrow 98$ |
| AntMaze Large Play | $0 \rightarrow 98$ | $0 \rightarrow 93$ | $86 \rightarrow 91$ |
| AntMaze Large Diverse | $0 \rightarrow 96$ | $0 \rightarrow 94$ | $85 \rightarrow 96$ |

flow-matching policy with a Q-guided prior, maintains the expressivity needed to represent diverse modes while achieving high sample efficiency during online adaptation.

- **Offline-to-Online Efficiency:** Ours starts with a significantly higher offline performance in most tasks (e.g., 51% in *Cube Double Play*) compared to the zero-start of SAC. This confirms that our "Golden Start" effectively anchors the policy in high-value regions from the beginning, while the entropy regularization ensures stable and principled exploration to reach near-perfect success rates (99%-100%) during fine-tuning.

## I  FUTURE WORK

While our method demonstrates improvements in continuous control tasks through Q-guided priors and entropy control, several promising directions remain for future research:

- **Unsupervised Skill Discovery:** We plan to extend the "Golden Start" concept to the domain of unsupervised skill discovery Eysenbach et al. (2018); Park et al. (2023); Zhang et al. (2025). By leveraging the expressive power of flow-matching policies, we aim to discover diverse and reusable primitive skills from unlabeled offline data without manual reward engineering. The learned priors could potentially serve as a structured latent space for skill induction and autonomous exploration.

- **VLA Models:** Integrating our framework into high-dimensional, multimodal VLA architectures (such as Black et al. (2024a); Intelligence et al. (2025)) is a priority. Preliminary investigations suggest that while data curation remains a bottleneck in VLA settings, the ability of our method to model complex, multi-modal distributions could significantly enhance the performance of embodied agents in tasks involving long-horizon reasoning and language instructions.

- **Generalization to Discrete Action Spaces:** Although the current implementation focuses on continuous control, the underlying principle of value-aligned generative priors is also applicable to discrete domains. We intend to explore the adaptation of flow-matching distillation to discrete action spaces, which could broaden the applicability of our method to a wider range of decision-making tasks, such as strategic games or combinatorial optimization Xu et al. (2019); Zhang et al. (2023).

