# OpenReview forum: "GoldenStart: Q-Guided Priors and Entropy Control for Distilling Flow Policies"
_ICLR.cc/2026/Conference — ICLR 2026 Poster_

### Official Review · Reviewer_ndGM · 2025-10-26

**Soundness:** 2
**Presentation:** 3
**Contribution:** 2
**Rating:** 2
**Confidence:** 4

**Summary:**

The authors introduce a golden start method for policy distillation with Q-guided priors. In addition, they allow the distiller to output a stochastic distribution based on entropy regularization, that allows the policy to explore beyond what has been learnt from offline data. The authors claim that they provide a new state-of-the-art.

**Strengths:**

The paper is nicely written and easy to follow. There are many comparisons with baselines (but see below). Combining a golden start with entropy regularization seems to be an important move forward.

**Weaknesses:**

It looks like there is an important typo in Table 1. The results claimed are not as good as I initially thought: In the 4 first tasks, the new method outperforms the previous ones only in 1 case. But a closer look reveals that this is a typo, as ReBRAC has a performance of 91, above the new method. Therefore, the new method does not show any benefit in the locomotion tasks.

Further, a closer look at the D4RL environments reveals that the improvement in performance is only marginal with respect to FQL, despite having a higher computational cost.

Second, the mathematics presented in the appendix are kind of trivial. Showing that under an updated policy that considers the highest values of a learned Q value there is going to be an improvement in reward is a trivial fact. I am not sure what the scope of Theorem 1 is beyond showing an expected result, which is so by construction.

Finally, the introduction of the entropy bonus term follows standard approaches in previous RL literature (Haarnoja, icml, 2018, quoted by the authors). From this addition it is totally expected that exploration would be improved. However, no comparison with continuous optimal controllers, such as SAC, are provided.

**Questions:**

What does the math in the Appendix reveal that is novel?

What would be the performance of the new distillation method against SAC, let us say, in locomotion and maze tasks?

Typos: Line 366 “achieves”, and Line 483 “establishes”.

---

> ### Author Response · Authors · 2025-11-16
> **Regarding W1, W2, Q1 and Q3**
>
> We sincerely thank you for your detailed review and insightful feedback! We appreciate the time you have taken to evaluate our work and we hope to address your concerns!
>
> ## Regarding W1: Performance on D4RL Locomotion and Comparison with FQL
>
> 1.  **On the "Typo" in Table 1 (Locomotion Tasks):** Thank you for your careful reading of our results. We would like to respectfully clarify that there is no typo in Table 1. All of our experimental data presented is accurate and reproducible.
>     We suspect your confusion may arise from comparing the ReBRAC results between Table 1 (offline setting) and Table 2 (offline-to-online setting, the value on the left of the arrow). These values are adopted from an original paper [1], which used different hyperparameters for the two settings (e.g., the behavior cloning weight is typically set lower for the online fine-tuning setting). Therefore, the values are not the same. Still, thanks for the comment. In our revised manuscript, we will add a clear note to explain this discrepancy.
>     While ReBRAC does perform well on the "medium" locomotion task in D4RL (a phenomenon also observed in [1, 2]), it is not good on the "large" task. We suspect this is because the "medium" map is more open (Let's say, it has more open space), making it easier to go out-of-distribution during evaluation. In such cases, a method with a harder behavior constraint (like ReBRAC) might yield better results.
>
> 2.  **On the "Marginal Improvement" over FQL:** We appreciate your observation! Our method is built upon FQL. As you said, our method did not bring a breakthrough improvement in D4RL locomotion tasks. However, we believe that achieving stable improvements over a strong baseline is also worth presenting [3]. Besides, in other challenging environments, our method surely achieved significant gains (such as Cube Double Play (offline setting) and Puzzle-4x4 Play (offline-to-online setting)). These results highlight our method's advantages in learning a superior starting policy and exploring more effectively, which are the core claims of our paper.
>
> ## Regarding W2 & Q1: Novelty of the Theoretical Analysis
>
> Thank you for your keen observation regarding the proof in the appendix. We are glad you found the proof straightforward and agree with its logic. We want to clarify that we did not intend for this theorem to be the novel contribution of our paper, which is why it was placed in the appendix. Our goal was not to introduce a new, complex mathematical framework, but rather to use the language of mathematics to provide a formal explanation for why our method yields improvements and to enhance the algorithm's theoretical rigor. We believe this validation, even if "by construction," is still a necessary component of our work. Still, we are open to any suggestions you may have to make this section more rigorous and impactful!
>
> ## Regarding Typos
>
> Thank you for catching these errors! We have corrected them in the manuscript to read "GS-flow achieves" and "GS-flow establishes".
>
> Due to space limitations, we will supplement W3 & Q2 below. Our apologies for the inconvenience.
>
> ---
>
> [1] Flow Q-learning
>
> [2] IDQL: Implicit Q-Learning as an Actor-Critic Method with Diffusion Policies
>
> [3] SCORE REGULARIZED POLICY OPTIMIZATION THROUGH DIFFUSION BEHAVIOR

---

> ### Author Response · Authors · 2025-11-16
> **Regarding W3 & Q2**
>
> Due to space limitations, we supplement W3 & Q2 additionally. Our apologies for the inconvenience. Again, we sincerely thank you for your detailed review and insightful feedback! We appreciate the time you have taken to evaluate our work and we hope to address your concerns!
>
> ## Regarding W3 & Q2: Entropy Bonus and Comparison with SAC
>
> Thanks for your suggestion. Comparing our entropy-based method to SAC [5] is indeed essential. We regret that we truly compared with a SAC-based algorithm, but it was not direct and clear for readers. One of our primary baselines, RLPD [4], is an excellent algorithm explicitly built upon and designed to improve SAC for utilizing offline data. The RLPD uses the same entropy as SAC, but has more stable and efficient performance. Our existing results in Table 2 (offline-to-online) already show that our method outperforms RLPD (SAC-based approach) on complex tasks (e.g., Cube Double Play) and matches its top-tier performance on exploration tasks (e.g., Puzzle-4x4 Play).
>
> We now realize this link was not explicitly written in our paper. In the final revision, we will clearly state the relationship between RLPD and SAC in the experimental setup (Page 7, Line 359) to make this comparison direct and clear for the reader.
>
> To fully resolve your concerns, we have conducted new experiments directly comparing our method against the original SAC on the OGBench and D4RL AntMaze tasks. (Note: SAC was run for 3 random seeds).
>
> | Task | SAC (Offline $\rightarrow$ Online) | RLPD (Offline $\rightarrow$ Online) | Ours (Offline $\rightarrow$ Online) |
> | :--- | :--- | :--- | :--- |
> | HumanoidMaze Medium | 0 $\rightarrow$ 2 | 0 $\rightarrow$ 84 | 5 $\rightarrow$ 67 |
> | AntSoccer Arena | 0 $\rightarrow$ 0 | 0 $\rightarrow$ 0 | 46 $\rightarrow$ 77 |
> | Cube Double Play | 0 $\rightarrow$ 0 | 0 $\rightarrow$ 0 | 51 $\rightarrow$ 99 |
> | Scene Play | 0 $\rightarrow$ 100 | 0 $\rightarrow$ 100 | 88 $\rightarrow$ 100 |
> | Puzzle-4x4 Play | 0 $\rightarrow$ 0 | 0 $\rightarrow$ 100 | 17 $\rightarrow$ 100 |
> | AntMaze U-Maze | 0 $\rightarrow$ 100 | 0 $\rightarrow$ 98 | 100 $\rightarrow$ 100 |
> | AntMaze U-Maze Diverse | 0 $\rightarrow$ 100 | 0 $\rightarrow$ 94 | 93 $\rightarrow$ 98 |
> | AntMaze Medium Play | 0 $\rightarrow$ 98 | 0 $\rightarrow$ 98 | 77 $\rightarrow$ 98 |
> | AntMaze Medium Diverse | 0 $\rightarrow$ 98 | 0 $\rightarrow$ 97 | 76 $\rightarrow$ 98 |
> | AntMaze Large Play | 0 $\rightarrow$ 98 | 0 $\rightarrow$ 93 | 86 $\rightarrow$ 91 |
> | AntMaze Large Diverse | 0 $\rightarrow$ 96 | 0 $\rightarrow$ 94 | 85 $\rightarrow$ 96 |
>
> As the table shows, SAC struggles significantly in the OGBench tasks. We attribute this to the multi-modal action spaces and sparse rewards, where SAC cannot effectively leverage the provided offline data. Our method achieves far superior results on the most complex tasks (e.g., AntSoccer, Cube Double Play).
>
> We thank you again for your time and constructive criticism. Your feedback has been invaluable in helping us strengthen our paper. We hope that these clarifications and new results have fully addressed your concerns!
>
> ---
>
> [4] Efficient Online Reinforcement Learning with Offline Data
>
> [5] Soft Actor-Critic: Off-Policy Maximum Entropy Deep Reinforcement Learning with a Stochastic Actor

---

> > ### Comment · Reviewer_ndGM · 2025-11-17
> >
> > I thank the authors for the detailed response. In particular the comparison with SAC is important and supports the author's claims.
> >
> > My comment regarding Table 1 was simply that the bold (winning performance) was attributed to "Ours" and not to ReBRAC in the AntMaze Large Navigate. I believe that this is still a typo.
> >
> > I still think that the paper does not have substantive theory that supports the main claims, and the work relies mostly on heuristic approximations and confirmatory empirical observations. For instance, there is no proof that the whole optimization problem will work in discrete state-action spaces. In this case a full theory with proofs of convergence could be attainable.
> >
> > Anyways, I will raise my final Rating, and I thank the authors for their responses.

---

> > > ### Author Response · Authors · 2025-11-17
> > >
> > > Thank you very much for your timely reply!
> > >
> > > We are very pleased to learn that our supplementary experiments regarding SAC have earned your acknowledgment. We confirm that we will add these results and the related discussion to the final version of the paper.
> > >
> > > At the same time, thank you very much for pointing out the typo in Table 1 again. You are correct; the bold attribution for AntMaze Large Navigate was indeed wrong. We will correct this error immediately in the final submission.
> > >
> > > Regarding the theoretical aspect, we fully understand your perspective. We highly value your advice and we are actively conducting a more in-depth analysis.
> > >
> > > Finally, our team sincerely thanks you for your decision to raise the score. Your constructive feedback and final support mean a great deal to our work.

---

> > > ### Author Response · Authors · 2025-11-28
> > >
> > > We sincerely thank the reviewer for a more positive reassessment of our work, and for raising the rating. And we acknowledge your concern about the theory.
> > >
> > > Here, we provide a supplementary theoretical analysis from the perspective of 'Amortized Optimization' to explain the 'Expected Performance Improvement'.
> > >
> > > ## Amortized Optimization Bound
> > > We view our method as an instance of Amortized Optimization (as mentioned by review NLQk). Here, the computationally costly test-time selection procedure (searching for the best noise) is "amortized" into a learned generative model (CVAE). We derive a performance bound showing that our amortized policy inherits the improvements of Best-of-$N$ search, minus a reconstruction error term controlled by the CVAE objective.
> > >
> > > **1. Oracle Policy via Rejection Sampling** Consider an idealized oracle that performs Best-of-$N$ sampling at test time. For a state $s$, it draws $N$ noise vectors $\\{x_i\\}\_{i=1}^N \sim \mathcal{N}(0,I)$ and selects the optimal one:
> > > $x^{\ast}\_{N}=\arg\max\_{x_i} Q(s, \pi(s, x\_i))$.
> > > By standard order-statistics results, the expected return of this oracle is strictly superior to a single random sample (the baseline):
> > > $$J\_{\text{oracle}} = \mathbb{E}[Q(s,\pi(s,x^{*}\_{N}))] \ge \mathbb{E}[Q(s,\pi(s,x))] = J\_{\text{base}}.$$
> > > We define this strictly non-negative gain as $\Delta_{\text{select}} = J_{\text{oracle}} - J_{\text{base}} \ge 0$.
> > >
> > > **2. Amortizing the Oracle via CVAE** To bypass the computational cost of running the oracle during online inference, we train a CVAE to approximate the distribution of the oracle-selected noise vectors $x^{\ast}\_{N}$. The training minimizes:$$\mathcal{L}\_{\text{VAE}} = \mathcal{L}\_{\text{recon}} + \lambda \mathcal{L}\_{\text{KL}},$$where the reconstruction error is defined as $\epsilon\_{\text{vae}} = \mathbb{E}\|x^{*}\_{N} - \hat{x}\|^2$, with $\hat{x}\sim p\_{\xi}(x|s)$ sampled from the learned CVAE prior.
> > >
> > > **3. Performance Bound** Let $V(x) = Q(s, \pi(s, x))$ be the value associated with a noise vector $x$. The expected return of our amortized policy is $J\_{\text{amortized}} = \mathbb{E}\_{\hat{x}\sim p\_\xi}[V(\hat{x})]$.We assume $V(x)$ is $L$-Lipschitz continuous with respect to the noise vector $x$, i.e., $|V(x) - V(y)| \le L \|x - y\|$.The performance gap between the oracle and our policy can be bounded as:$$J\_{\text{oracle}} - J\_{\text{amortized}} = \mathbb{E}[V(x^{\ast}\_N) - V(\hat{x})] \le L \cdot \mathbb{E}\|x^{\ast}\_N - \hat{x}\|.$$Applying Jensen's inequality ($\mathbb{E}[X] \le \sqrt{\mathbb{E}[X^2]}$), we have:$$J\_{\text{oracle}} - J\_{\text{amortized}} \le L\sqrt{\mathbb{E}\|x^{\ast}\_N - \hat{x}\|^2} = L\sqrt{\epsilon\_{\text{vae}}}.$$Substituting $J\_{\text{oracle}} = J\_{\text{base}} + \Delta\_{\text{select}}$, we obtain the final lower bound:$$J\_{\text{amortized}} \ge J\_{\text{base}} + \underbrace{\Delta\_{\text{select}}}\_{\text{Selection Gain}} - \underbrace{L\sqrt{\epsilon\_{\text{vae}}}}\_{\text{Amortization Gap}}.$$
> > >
> > >
> > > ## Interpretation and Alignment with Empirical Findings
> > > The derived bound $J_{\text{amortized}} \ge J_{\text{base}} + \Delta_{\text{select}} - L\sqrt{\epsilon_{\text{vae}}}$ offers a theoretical explanation for several empirical phenomena observed in our experiments:
> > >
> > > 1. **Effect of $N$ ($\Delta_{\text{select}}$):** The term $\Delta_{\text{select}}$ represents the potential gain from the Best-of-$N$ search. As $N$ increases, the expected maximum of the samples increases, raising $\Delta_{\text{select}}$. This aligns perfectly with Figure 6(a), where we observe that performance improves consistently as we increase the number of candidates $N$ used to train the prior.
> > >
> > > 2. **Robustness to Q-landscape ($L$):** The Lipschitz constant $L$ reflects the smoothness of the value landscape. Even in environments with sharp transitions (large $L$), such as the Multi-Crescent toy environment we designed, our method maintains performance gains. This suggests that while a smaller $L$ tightens the bound, the selection gain $\Delta_{\text{select}}$ is often sufficient to overcome the approximation gap even in complex landscapes.
> > >
> > > 3. **Importance of Reconstruction ($\epsilon_{\text{vae}}$):** The bound explicitly shows that minimizing the reconstruction error $\epsilon_{\text{vae}}$ maximizes the lower bound of the return. This corroborates our response to Reviewer frzm and the results in Table 3, where we found that the primary variation in VAE loss comes from the reconstruction term. As $\epsilon_{\text{vae}}$ decreases, the policy's performance improves, validating that better approximation of the oracle directly translates to higher returns.
> > >
> > > We believe that this additional analysis strengthens the theoretical grounding of our work. We sincerely appreciate your constructive feedback!

---

### Official Review · Reviewer_NLQk · 2025-10-29

**Soundness:** 2
**Presentation:** 3
**Contribution:** 2
**Rating:** 4
**Confidence:** 3

**Summary:**

The main contribution of this paper is a new method to learn a “golden prior” noise distribution for flow matching, that produces higher-valued actions than the initial Gaussian distribution. This prior is learned by sampling many noise vectors, mapping them to actions through flow matching, and then training a conditional VAE to generate only the highest-valued of the noise vectors. This transformed distribution is then used to seed the flow-matching policy for action selection. The starting point for the broader algorithm is FQL, and this modification, along with adding stochasticity and entropy-regularization to the policy, improve the performance of the final algorithm.

**Strengths:**

The idea of Golden priors is an interesting one, and there are not many instantiations of it. As this paper points out, directly optimizing flow-matching policies using RL has many challenges that this method avoids. Advantage Noise Selection is a reasonable method for creating higher-valued noise targets. The experimental results are thorough, and the crescent experiment is a very good demonstration of the method doing what it claims. The method achieves quite good performance on the extensive benchmark suite.

**Weaknesses:**

I would describe the instantiation of Advantage Noise Selection as amortizing the rejection-sampling process. I think a stronger argument could have been made for, and more time spent on, why we expect an advantage of this method over rejection sampling (IFQL is the included reference for this) — why does this lead to better performance when it is doing something pretty similar? The 2x speedup I think is not strength enough alone given the added complexity.

The entropy regularization is empirically helpful, and good to have tested, but I would argue not very surprising.

I think the theory section has some weaknesses. First, it relies on the prior actually producing better initializations, and Lemma 2 is not quite rigorous. And second, is that implicit in Theorem 1 is that sampling a higher-Q-estimate action is always than sampling a lower-Q-estimate action. When this in fact depends on whether your Q function can be trusted, which in off-policy RL is one of the main difficulties, and why we encourage sticking to the prior. So I don’t think this proof is wrong, but also doesn’t provide much supporting evidence for the model’s effectiveness. As a minor thing, I think the theory should be mentioned in the main text, as I was surprised to see it when I scrolled down.

Another minor thing, L_distill is confusingly included twice meaning two different things, Eq 10 and Eq 2.

**Questions:**

Mainly, I would like to better understand the comparison of IFQL and other rejection-sampling methods with yours. First, what “N” was used for rejection sampling in IFQL? Second, should we expect Advantage Noise Estimation to be better than rejection sampling, or just faster?

---

> ### Author Response · Authors · 2025-11-19
> **Regarding W1, Q1 and Q2**
>
> We sincerely appreciate your time! Thank you for the constructive feedback and the insightful questions regarding our motivation, comparisons with rejection sampling, and theoretical rigor. To address your comments, we respond to each point below.
>
> ### **Response to W1: Motivation and Efficiency**
>
> We appreciate the reviewer’s perspective on the trade-off between performance and complexity. Regarding the concern about whether the computational cost justifies the gains, we provide a detailed comparison in **Response to Q2** demonstrating that our method is fundamentally **better** in terms of stability and safety, not just faster.
>
> Regarding the motivation, we clarify that our inspiration stemmed not from IFQL, but from recent discussions in [1]. It has been observed that sampling noise multiple times and filtering actions based on Q-values improves online exploration stability. However, performing this repeated sampling during online interaction introduces severe computational overhead.
>
> This insight led us to train a "prior model" to automate and "amortize" this sampling process—resulting in our VAE design. We emphasize that even if the inference speedup is only "2x", this difference becomes critical when scaled over **1 million online interaction steps**. In the context of real-time robotic control, reducing this cumulative latency is a significant contribution.
>
> ### **Response to Q1: The value of $N$ used in IFQL**
>
> In all benchmarks reported in the paper, the value of $N$ for IFQL is fixed at **32**. We did not tune the parameter $N$ for IFQL.
>
> ### **Response to Q2: Comparison to Rejection Sampling (RS)**
>
> We demonstrate that our method (GS-flow) is **better**, not merely faster, than Rejection Sampling (RS). While RS relies on stochastic filtering at inference time, GS-flow actively optimizes the generative process during training. We highlight the superiority of our approach regarding **stability**, **robustness against overestimation**, and **safer distillation**.
>
> **1. Superior Stability and Precision**
> Our method achieves significantly lower variance compared to IFQL. As shown in the table below, IFQL relies on probabilistic sampling from a standard Gaussian distribution. This makes it difficult to guarantee high-quality actions in every iteration, leading to higher variance. In contrast, ours learns a structured Q-guided prior via a CVAE. As the CVAE loss minimizes, the prior fits the distribution of high-value actions, allowing for stable generation without redundant sampling.
>
> | Method | IFQL ($N=24$) | IFQL ($N=28$) | IFQL ($N=32$) | **Ours** |
> | :- | :-: | :-: | :-: | :-: |
> | **Reward (Avg)** | 7.6 | 8.1 | 8.0 | **9.2** |
> | **Reward (Std)** | 1.5 | 1.6 | 1.5 | **0.2** |
>
> **2. Robustness to Q-Value Overestimation**
> The data above shows that for IFQL, increasing $N$ from 28 to 32 does not yield further improvements (8.1 $\rightarrow$ 8.0). In RS, increasing $N$ increases the probability of selecting Out-of-Distribution (OOD) actions where the Critic network erroneously predicts high Q-values (overestimation).
>
> In contrast, our Advantage Noise Selection is **anchored** to the data distribution. The target "advantage noises" are derived exclusively from the **Teacher Policy**, which is trained via Behavioral Cloning and strictly constrained to the offline dataset's support. Consequently, our student policy learns to reproduce optimal modes that are **verified within the data distribution.**
>
> **3. Further Analysis on Robustness**
> To isolate the benefit of our distillation framework, we compared GS-flow against an FQL agent enhanced with Rejection Sampling ($N=32$) on the `Cube-Double` task, where Q-overestimation is severe.
>
> | Method | IFQL | FQL | FQL-RS | **Ours** |
> | :- | :-: | :-: | :-: | :-: |
> | **Success Rate (Avg)** | 9 | 36 | 37 | **51** |
> | **Success Rate (Std)** | **5** | 6 | 7 | 6 |
>
> Even with RS added, FQL fails to match GS-flow (37 vs. 51). GS-flow is structurally superior from two perspectives:
>
> * **3.1 Gradient Alignment:** In FQL, the distillation loss pulls the policy towards the teacher's *average* behavior (conditioned on standard noise), while the Q-loss pulls it towards the *mode* (highest value). These directions often conflict. Since RS merely filters outputs at inference time, it cannot mitigate this training conflict. In GS-flow, the distillation target is pre-selected for high value. Thus, the distillation and Q-maximization objectives are **aligned**, pushing the policy consistently towards the optimal mode.
> * **3.2 Input Manifold Constraint:** FQL allows the optimizer to warp the mapping from *any* random noise to high-Q actions, risking overfitting to noise vectors that trigger critic overestimation. GS-flow constrains the student's input to the **learned prior (CVAE)**. This restricts the student to a "safe" latent manifold known to generate valid, in-distribution actions.
>
> (Continued in the next comment due to space limits. Apologies for the inconvenience.)

---

> ### Author Response · Authors · 2025-11-19
> **Regarding W2, W3 and W4**
>
> ### **Response to W2: Entropy Control**
>
> We agree that the Entropy Regulator is a well-established paradigm. However, we believe **how we enable entropy control within the distillation framework** is a valuable contribution addressing a core limitation in generative policies.
>
> The inherent challenge in Diffusion/Flow Policies is the **intractability of the policy's exact entropy** (due to the complex generative process). Existing methods resort to approximations: distribution fitting via GMM [2], entropy lower bounds [3], or numerical integration [4].
>
> To solve this within our distillation framework, we replaced the traditional **"point-to-point" deterministic mapping** with an explicit **"point-to-distribution" output** (mean $\mu_{\varphi}$ and standard deviation $\sigma_{\varphi}$). Given a noise input, the model outputs a Gaussian distribution, allowing us to directly employ a standard SAC-style objective. This design enables **principled online exploration**—a major deficit in standard FQL—in a lightweight manner, complementary to our Q-Guided Prior.
>
> ### **Response to W3: Theoretical Analysis**
>
> We thank the reviewer for the analysis. Following your suggestion, we have **added a reference in Section 3.2 explicitly pointing to the theoretical analysis in the Appendix.**
>
> We agree that the original Lemma 2 relied on assumptions regarding error minimization. To improve rigorousness, **we have revised Lemma 2 into "Proposition 1: Sufficient Condition for Prior Improvement" in the updated manuscript.**
> Instead of asserting unconditional superiority, the new Proposition 1 establishes the sufficient conditions for our method to outperform the baseline. Specifically, we prove that the learned prior $p_{adv}$ improves upon the initial prior $p_0$ if the sum of the optimization error ($\epsilon_{opt}$) and statistical error ($\epsilon_{stat}$) is less than the fixed baseline error ($C_0$):
>
> $$\epsilon_{opt} + \epsilon_{stat} < C_0$$
>
> We further clarified that while the VAE objective (ELBO) is not $W_1$, minimizing the VAE loss maximizes the likelihood of the advantage samples, effectively driving the learned distribution closer to the target and reducing $\epsilon_{opt}$. This revision shifts the claim to a "theoretical guarantee under achievable conditions," which is mathematically sound.
>
> ### **Response to W4: Notation Clarification**
>
> We sincerely apologize for the confusion regarding the symbol ${L}_{Distill}$. We have corrected this in the revision:
>
> 1.  **Section 2.2 (FQL Baseline):** We have renamed the overall loss in Eq. 2 to ${L}_{FQL}$ to clearly denote the composite loss (Q-learning + Distillation) of the baseline.
>
> 2. **Section 3.3 (GS-flow):** We retained the term ${L_{Distill}}$ in Eq.10 but explicitly renamed it to $L_{\text{L2-Distill}}$ to refer strictly to the L2 distance term:
>
>      $L_\text{L2-Distill} = \mathbb{E}[ \|| \mu_{\varphi}(s, x_{adv} ) - a_{teacher} \||^{2} ]$
>
> This ensures clear distinction between the baseline's total loss and our specific distillation objective.
>
>
> Thank you again for the time and effort dedicated to reviewing our paper! We hope that our response has effectively addressed your concerns. Our team remains available for any further discussions during the rebuttal period.
>
> ---
> **References**
>
> [1] Efficient Online Reinforcement Learning for Diffusion Policy
>
> [2] Diffusion Actor-Critic with Entropy Regulator
>
> [3] DIME: Diffusion-Based Maximum Entropy Reinforcement Learning
>
> [4] Maximum Entropy Reinforcement Learning with Diffusion Policy

---

> > ### Comment · Reviewer_NLQk · 2025-11-21
> > **Concerns mostly addressed**
> >
> > My main concern relating to the comparison between this method and rejection-sampling was addressed. Thank you for the additional experiments and explanation.
> >
> > I remain unconvinced by the description of entropy. First, as I understand it the entropy of the policy is still an estimate, because the true policy distribution $H(\pi(a|s))$ also integrates over the flow-matching initial Gaussian, not just the SAC Gaussian. I see it helps but I think it's just the same reason SAC is better than TD3.
> >
> > But I still think this is a valuable contribution and have changed my rating. Thank you for the minor presentation improvements as well.

---

### Official Review · Reviewer_hfGV · 2025-10-30

**Soundness:** 4
**Presentation:** 4
**Contribution:** 3
**Rating:** 8
**Confidence:** 3

**Summary:**

This paper introduces GoldenStart, an improvement of previous existing methods for offline and offline-to-online reinforcement learning tasks, that requires to model and learn complex distributions.

Method:

This work builds on Flow Q-learning (FQL), a method based on the actor-critic structure. FQL is based on flow matching: the goal is to train a distilled (one-step) state-conditional flow student network to output actions that (i) maximize the value function (ii) while minimizing a distillation loss against a flow teacher. The flow teacher is trained to do behavioral cloning on an offline dataset D using multiple timesteps (unlike the student), akin to the standard flow matching method. (i) encourages exploitation and (ii) encourages staying close to the collected high reward dataset D.
On top of this, GoldenStart introduces 2 main contributions:
-  First, instead of using gaussian noise as prior,  they learn a Q value guided prior model as a state-conditional VAE.
     Such a conditional VAE is trained in 2 steps:
          - First, they generate "advantage noises". Given a state and N_cand initial gaussian noises, they generate actions using the student flow network, and they keep the action with the highest value function,
           - Second, they train a state-conditional VAE that learns to generate the advantage the distribution of advatange noises from a gaussian noise.
- Second, they train the student flow to predict a mean and a variance. Why? Because in addition to the value maximization term and the distillation term of FQL, that were already present in FQL, they add an entropy term. This encourages exploration in addition to exploitation.

Experiments:

They introduce a new toy task called crescent task,  and they do ablation studies to illustrate the added value of each of their contributions in terms of exptected return.
They also use the same experimental setups as in FQL (i.e. OGBenchmark, AntMaze, Visual envs ), and the average expected return (over different subtasks) is higher with GoldenStart than with FQL and other baseline methods .

**Strengths:**

- The clarity of the paper, the explanations, the contributions and evidence supporting them
- The introduction of the crescent toy task with the 2D visualisations, as well as the ablation studies in the experiments for crescent for each of the contributions (prior / entropy), which helps in understanding,
- The simplicity of the introduced changes
- The exhaustive set of experiments (as in FQL)

**Weaknesses:**

- The additional computational cost for learning the VAE prior distribution

**Questions:**

- The teacher flow network trained on the offline dataset using gaussian noise as a starting point, but in the text you do inference from it using the distribution of x_advantage as a starting distribution. Could you explain how the teacher adapts to the distributional shift of this initial distribution?
- Correct me if I am wrong, but it seems to me that the conditional VAE is learns a "higher return actions distribution" than the student flow network (because of the argmax in the Q value over N_candidates, and the absence of the entropy regularization term) ? If yes, could you show the expected return over some tasks in the OGBench using the advantage noise selection module only?
- Why do you need to predefine an entropy target H_{target}? How do you choose it in your exps? Does it depend on the task?
- Do you have some results that could show that GoldenStart is doing principled exploration that FQL is not doing? E.g. discovering some actions that FQL does not discover, and that are not in the offline dataset D ? (It is difficult to infer from the expected return only that GoldenStart allows for better exploration compared to FQL).

---

> ### Author Response · Authors · 2025-11-20
> **Regarding Q1 and Q2**
>
> We sincerely thank you for your support and your recognition of our paper's clarity, the design of the Crescent task, and the comprehensiveness of our experiments. We also value the opportunity to address your inquiries and engage in this discussion!
>
> ## Response to Q1: Potential Distributional Shift
>
> Your concern regarding the potential distributional shift is very reasonable. However, we argue that in our current experimental tasks, the VAE fitting is sufficient, and the distributional shift does not negatively impact the teacher network.
>
> **1. Theoretical Bound**
> The VAE training data is sampled and selected directly from the standard Gaussian distribution. This means the VAE learns to approximate a high-value sub-region strictly within the teacher's valid input domain. Therefore, theoretically, the VAE's natural outputs fall well within the teacher's effective input space.
>
> **2. Empirical Verification**
> We verified this in our preliminary experiments by explicitly clipping the VAE outputs to the range $[-1, 1]$. We observed that this constraint had a **negligible impact** on the final performance. This empirical evidence confirms that the learned outputs naturally stay within bounds and do not cause Out-of-Distribution (OOD) issues.
>
> ## Response to Q2: Role of CVAE and Ablation Study
>
> We appreciate the insightful question regarding the role of the conditional VAE (CVAE) and the suggested ablation study. We would like to clarify the mechanism and explain why the proposed ablation is structurally difficult to implement, while offering a more rigorous comparison to validate our contribution.
>
> **1. Clarification on VAE's Role**
> The reviewer asks if the CVAE learns a "higher return action distribution." We wish to clarify that the CVAE is designed to predict a **coarse-grained** "advantage noise distribution" rather than the action distribution directly. We need the student model to learn based on its prior distribution.
>
> **2. Structural Dependency of the Modules**
> Regarding your request to test the Advantage Noise Selection module, we wish to argue that the Selection module and the CVAE constitute an **inseparable framework**.
>
> The Advantage Noise Selection is a only training-time mechanism to identify high-value noises to train CVAE. The CVAE is to provide a prior distribution for student model both in training time and inference time. However, without the CVAE during inference time, we would have no mechanism to generate noises but sampled from standard Gaussian distribution, which would be OOD of what we input to the student model during training time.
>
> **3. Validating Effectiveness via Rejection Sampling**
> We understand your suggestion is to prove the benefit of the VAE. To verify the effectiveness of our CVAE, we compared it against Rejection Sampling (RS) based on FQL.
>
> Under the FQL training framework, FQL-RS involves sampling multiple actions from a standard normal distribution and selecting the one with the highest Q-value. Although this mimics the behavior of our Advantage Noise Selection module, the critical difference is that the data filtered by rejection sampling cannot be used to train the student model. This is because, without the CVAE, we would not have access to the specific noise distribution needed for the student model during inference. Consequently, this baseline uses rejection sampling solely for critic updates and final evaluation.
>
>
> We compare our method against an FQL agent that samples $N$ actions from a Gaussian prior and selects the one with the highest Q-value during inference. We test on the **Puzzle-4x4** environment.
>
> | Method | FQL | FQL-RS | FQL-VAE (Ours) |
> | :--- | :---: | :---: | :---: |
> | **acc_avg** | 36 | 37 | **51** |
>
> This comparison demonstrates that simply filtering high Q-value actions during the evaluation process (Rejection Sampling) is insufficient. Our framework effectively learns and generalizes the high-value noise distribution, providing a superior prior that boosts distilling performance.
>
> (Continued in the next comment due to space limits. Apologies for the inconvenience.)

---

> ### Author Response · Authors · 2025-11-20
> **Regarding Q3 and Q4**
>
> ## Response to Q3: Configuration of Target Entropy $H_{target}$
>
> We thank the reviewer for their careful review of our method.
>
> 1. **Why predefined $H_{target}$?**
>     The use of $H_{target}$ is intended to implement **Automatic Entropy Adjustment** for the temperature parameter $\alpha$. By constraining the policy entropy against $H_{target}$, the algorithm dynamically adjusts $\alpha$. This allows the agent to automatically decrease $\alpha$ during the early stages of training (when the policy is random and entropy is high) to accelerate learning, and increase $\alpha$ when the policy converges (when entropy is low) to maintain exploration. This mechanism effectively eliminates the need for manual hyperparameter tuning, consistent with improvements seen in Soft Actor-Critic (SAC)[1].
>
> 2. **How to set the value?**
>     In our experiments, we followed the common heuristic setting of $H_{target} = -0.5 \cdot \dim(\mathcal{A})$ (in RLPD[2] Appendix B.2) for the majority of tasks. If the task requires more exploration, we would increase the value.
>
> 3. **Does it depend on the task?**
>     We found that the optimal $H_{target}$ can be task-dependent. This is particularly evident in the OGBench Puzzle-4x4 task, which relies heavily on the model's exploration capability. Consequently, baselines without entropy control (like FQL) perform significantly worse than algorithms with entropy control (like RLPD) on this task. In this specific scenario, we increased the target entropy to $H_{target} = -0.1 \cdot \dim(\mathcal{A})$, which enabled the agent to achieve almost 100% success rate during online exploration.
>
> ## Response to Q4: Evidence of Exploration Capability
>
> We thank the reviewer for this insightful question regarding our exploration efficiency.
>
> 1. **Qualitative Analysis (Figure 5):**
>     Figure 5 provides evidence of this exploration capability. In the online setting, our algorithm discovered regions that FQL missed. Specifically, in **Figures 5(d) and 5(e)**, the two distributions with the highest rewards (located at the bottom-left and top-right, respectively) are both absent from the offline dataset. While FQL only identifies the bottom-left distribution, our algorithm successfully discovers both.
>
> 2. **Quantitative Validation (Puzzle 4x4):**
>     We further validated this on the Puzzle 4x4 task (averaged over 3 runs). In this environment (a 4x4 matrix), the goal is to turn off all lights (set the matrix to 0) by toggling states. We compared the `max_button_state` metric, defined as the maximum value of the matrix mean observed during the entire online exploration process.
>
> | Method | FQL | Ours |
> | :--- | :---: | :---: |
> | **max_button_state** | 0.34 | **0.53** |
>
> We found that our algorithm achieved a `max_button_state` of **0.53**, which is significantly higher than FQL's **0.34** (these maximums primarily occurred during the early stages of online exploration). We suppose this demonstrates that our algorithm indeed reached states that FQL failed to visit.
>
>
>
>
> Finally, we thank you again for your time and insightful comments. We hope that these clarifications and new results have fully addressed your concerns!
>
> ---
>
> [1] Soft actor-critic: Off-policy maximum entropy deep reinforcement learning with a stochastic actor.
>
> [2] Efficient online reinforcement learning with offline data.

---

### Official Review · Reviewer_frzm · 2025-11-01

**Soundness:** 3
**Presentation:** 3
**Contribution:** 3
**Rating:** 8
**Confidence:** 3

**Summary:**

This paper proposes GoldenStart (GS-flow), a novel framework for distilling flow-matching policies in reinforcement learning. The authors identify two key limitations of existing one-step distilled flow policies:

(1) the uninformed Gaussian prior that ignores the value landscape, and (2) the deterministic mapping that prevents controllable exploration.

To address these issues, GS-flow introduces two main innovations:

(1) Q-Guided Prior Learning, where a conditional VAE learns a state-conditioned prior that biases the initial noise toward high-Q regions, providing a “golden start” for generation; and
(2) Entropy-Regularized Distillation, which replaces deterministic action generation with a stochastic actor trained under an entropy-regularized loss, enabling adaptive exploration during online fine-tuning.
Extensive experiments on OGBench, D4RL AntMaze, and Visual RL benchmarks demonstrate that GS-flow consistently outperforms prior state-of-the-art methods (e.g., FQL, IFQL), particularly in multi-modal and exploration-heavy tasks such as Cube Double and Puzzle-4×4. The method preserves single-step inference efficiency while achieving significant gains in both offline and online performance.

**Strengths:**

The paper precisely identifies two previously overlooked weaknesses in existing one-step distilled flow policies — the use of an uninformed Gaussian prior and the absence of controllable stochasticity — and builds upon them to formulate a coherent research question and motivation.

The work introduces a conditional VAE that learns a state-conditioned high-Q prior, transforming the random starting point in generative inference into a structured, value-aligned initialization, thereby providing an effective “shortcut” for policy optimization.

The authors reformulate the student policy from a deterministic mapping into a distributional generator with entropy control, achieving a principled balance between exploration and exploitation and significantly enhancing online adaptability.

Across benchmarks such as OGBench, D4RL, and Visual RL, GS-flow consistently achieves state-of-the-art performance, demonstrating superior sample efficiency and exploration capability while maintaining single-step inference speed. Moreover, the ablation studies clearly disentangle and validate the independent contributions of the prior learning and entropy regularization modules.

**Weaknesses:**

1. The Advantage Noise Selection module relies on a hard argmax over Q-values to identify the optimal noise per state. While simple, this approach may amplify critic bias and reduce diversity in the learned prior. More robust alternatives such as soft advantage weighting or top-k filtering could potentially mitigate this brittleness.

2. While GS-flow demonstrates impressive efficiency within the family of flow-matching methods (notably compared to FQL and IFQL), it remains unclear whether this efficiency translates to practical advantages over lightweight Gaussian actor-critic approaches such as SAC or IQL. Given the additional complexity of the teacher–student architecture and prior learning modules, the overall time–performance trade-off might still be less favorable in real-world applications.

**Questions:**

1. The Multi-Crescent environment is an interesting design to expose Q-value overestimation, but given its continuous nonconvex reward surface, there might be potential for spurious high-Q regions or reward hacking when the critic is imperfect. Could the authors examine whether GS-flow’s Q-guided prior might amplify such artifacts? For instance, visualizing true reward vs. predicted Q over the state-action space or adding a calibration/consistency check (e.g., reward clipping or Q–r alignment) would strengthen the claim that the method is robust to reward misestimation.

2. The actor is trained with a composite objective combining distillation, Q-value maximization, and entropy regularization. Given the potentially conflicting gradients among these components, could the authors clarify whether they observed optimization instability or gradient interference during training?

3. Have the authors explored combining GS-flow with language- or vision-conditioned policies (e.g., VLA settings) to test generality beyond low-dimensional control?

4. Could the authors provide empirical insights into the interaction between entropy temperature α₂ and Q-guided prior learning — are they synergistic or competing during online fine-tuning?

---

> ### Author Response · Authors · 2025-11-22
> **Regarding Q1 and Q2**
>
> We are truly grateful for your positive assessment and encouraging feedback. We apologize for the delay in our response. We wanted to ensure our answers were supported by rigorous empirical evidence, which required time to carefully design and run additional ablation studies. Below, we provide detailed responses to your questions.
>
> # Response to Q1: Potential Q-Overestimation and Reward Hacking
>
> We appreciate this excellent point. We acknowledge that the Multi-Crescent environment is designed to expose overestimation risks. In fact, our existing **Appendix D (Figure 8)** demonstrates this risk: when the BC weight ($\alpha$) in FQL is set too low ($\alpha=1$), actions cluster in the "gap" of the crescent (reward hacking), whereas a higher weight ($\alpha=100$) prevents this.
>
> To rigorously examine whether GS-flow amplifies these artifacts, we conducted an additional sensitivity analysis on the BC weight $\alpha_1$ (Eq. 9). We measured the bias between the predicted $Q$ and true return $r$, specifically calculating `bias = Q - r` and `abs_bias = |Q - r|`.
>
> **Table 1: Analysis of Q-Overestimation and Robustness**
> | Model Setting | Ours ($\alpha_1=0$) | Ours ($\alpha_1=1$) | Ours ($\alpha_1=10$) | Ours ($\alpha_1=100$) | Ours w/o VAE ($\alpha_1=100$) |
> | :--- | :--- | :--- | :--- | :--- | :--- |
> | **Bias ($Q-r$)** | 10.0 | 7.0 | 1.1 | **0.4** | 0.3 |
> | **Abs Bias** **($ \|Q-r\| $)** | 10.0 | 7.0 | 1.1 | **0.4** | 0.3 |
>
>
> **Table 2: Ablation test for VAE**
> | Model Setting | Ours ($\alpha_1=100$) | Ours w/o VAE ($\alpha_1=100$) |
> | :--- | :--- | :--- |
> | **Return** | **9.3** | 7.0 |
>
> **Observations & Robustness:**
> 1.  **Confirmation of Overestimation:** The fact that `Bias` equals `Abs Bias` across all settings confirms that all estimation errors are **overestimations** (no underestimation), validating that the environment indeed encourages reward hacking.
> 2.  **Sensitivity to $\alpha_1$:** As expected, reducing the BC constraint ($\alpha_1 \to 0$) leads to severe overestimation (Bias reaches 10.0), causing the policy to exploit spurious high-Q regions.
> 3.  **Robustness of GS-flow:** Crucially, with a reasonable setting ($\alpha_1=100$), GS-flow exhibits only marginal overestimation compared to the baseline (0.4 vs. 0.3). However, the performance gain is significant (**Return 9.3 vs. 7.0**). This indicates that while minor overestimation exists, our Q-guided prior successfully directs the agent to legitimate high-value regions rather than "hacking" the reward function.
>
>
> # Response to Q2: Optimization Instability and Gradient Interference
>
> Thank you for raising this critical implementation detail. During development, we indeed encountered the challenge of balancing gradients between Distillation, Q-maximization, and Entropy regularization. We address this in two parts:
>
> 1.  **Q-Maximization vs. Entropy (Solved via Auto-tuning):**
>     To handle the conflict between exploitation (Q-max) and exploration (Entropy), we adopted the automatic entropy adjustment mechanism standard in SAC-based methods (Eq. 12 & 13). This allows the agent to dynamically adjust $\alpha_2$: exploring when entropy is below the target and exploiting when it is sufficient.
>     * *Note on Tuning:* While helpful, we admit that the target entropy hyperparameter ($H_{target}$) still requires task-specific tuning. For example, exploration-heavy tasks like **Puzzle-4x4** required a higher penalty ($H_{target} = -0.1 \times \dim(A)$) compared to standard tasks ($H_{target} = -0.5 \times \dim(A)$) to stabilize online performance.
>
> 2.  **Distillation vs. Q-Maximization (Managed via Scheduling):**
>     Balancing the BC term (distillation) and RL term is a known open challenge in offline RL (e.g., *ReBRAC, FQL*).
>     * **Offline Phase:** We found that using the default $\alpha$ from FQL as our $\alpha_1$ yields stable offline results.
>     * **Online Phase:** We observed that relaxing the conservation constraint is beneficial. We reduce $\alpha_1$ during online fine-tuning to lower the BC gradient weight and prioritize Q-maximization, allowing the agent to improve beyond the teacher's demonstration.
>
> (Continued in the next comment due to space limits. Apologies for the inconvenience.)

---

> ### Author Response · Authors · 2025-11-22
> **Regarding Q3 and Q4**
>
> # Response to Q3: Generalization to Vision-Language-Action (VLA) Models
>
> We are very glad you share this vision! We are currently actively working on integrating GS-flow into a VLA framework (specifically based on the **pi0** architecture).
>
> **Preliminary Findings:**
> Our initial investigations suggest that the primary bottleneck in VLA settings currently lies in **data quality and representation** rather than the expressivity of the flow-matching policy itself. While we do not yet have conclusive results to present in this revision, preliminary experiments indicate that the "Golden Start" concept could be particularly valuable for VLA tasks where multimodal distributions are common, but data curation remains the immediate hurdle. We plan to release these results in future work focused specifically on embodied AI.
>
>
>
> # Response to Q4: Interaction between Entropy Temperature $\alpha_2$ and Q-guided Prior
>
> We appreciate this insightful question. Based on our empirical observations, we find that $\alpha_2$ and the Q-guided prior are **synergistic** rather than competing during online fine-tuning.
>
> To demonstrate this, we conducted an ablation study on the Multi-Crescent environment. We initialized all agents from the same offline checkpoint (containing the learned Q-prior) and performed online fine-tuning for 100k steps. We varied the hyperparameter `target_entropy_multiplier` (denoted as `mult`). Specifically, the target entropy is calculated as: `target_entropy = - mult * action_dim` to automatically tune $\alpha_2$. Consequently, a larger `mult` results in a lower target entropy, thereby reducing the exploration incentive.
>
> We measured the Return, final $\alpha_2$, and VAE Loss.
>
> **Table 3: Interaction Analysis on Multi-Crescent Environment.** (Note: The observed differences in VAE Loss are primarily driven by the KL divergence component (kl_loss), while the reconstruction loss shows negligible variation.)
>
> | Setting | Target Entropy | Return | Final $\alpha_2$ | VAE Loss |
> | :--- | :--- | :--- | :--- | :--- |
> | **Ours (High Exp.)** | High (`mult=0.1`) | **14.0** | 3.6 | **0.090** |
> | **Ours (Med Exp.)** | Medium (`mult=0.5`) | 12.8 | 2.3 | 0.093 |
> | **Ours (Low Exp.)** | Low (`mult=0.75`) | 9.3 | 1.9 | 0.117 |
> | **Ours w/o Q-Prior** | High (`mult=0.1`) | 5.9 | 2.1 | - |
>
> **Analysis:**
>
> 1.  **Q-Prior as the Foundation:** The variant `w/o Q_prior` suffers a significant performance drop (14.0 $\to$ 5.9). This confirms that the Q-prior provides the distribution ensuring a valid performance lower bound and protecting the student policy from OOD inputs during early fine-tuning.
> 2.  **Entropy Drives Optimization:** Increasing the target entropy (`mult=0.75` $\to$ `0.1`) leads to a substantial increase in Return (9.3 $\to$ 14.0). While the Q-prior points to the general high-value region, the entropy term ($\alpha_2$) is essential for driving the local exploration needed to find the optimal solution within that region.
> 3.  **Mutual Benefit (Key Insight):** We observe a strong correlation where **higher entropy leads to lower VAE loss**.
>     * When exploration is constrained (Low Exporation), the policy tends to collapse into sharp, potentially suboptimal local modes. These sharp distributions are difficult for the VAE prior to model, resulting in a higher KL loss (0.117).
>     * When exploration is sufficient (High Exporation), $\alpha_2$ smooths the policy distribution. This **smoothing** effect prevents mode collapse and makes the high-value actions easier for the Prior to fit (lowest KL loss: 0.090). This proves that entropy regularization actively assists the prior learning process by maintaining a learnable distribution landscape.
>
>
> We hope these additional experiments and clarifications adequately address your questions. The suggested analyses have significantly strengthened the empirical analysis of our work. We are committed to incorporating these new results and detailed discussions into the final revision of the paper. Thank you again for your valuable time and support!

---

> > ### Comment · Reviewer_frzm · 2025-11-27
> >
> > Thank you for your response. My concerns have been addressed, and I will maintain my positive assessment.

---

### Comment · Area_Chair_6DVM · 2025-11-21
**Author-Reviewer Discussion**

Dear reviewers,

Please review the authors' response and adjust your rating accordingly. If you have any further questions, please discuss with the authors further.

AC

---

### Author Response · Authors · 2025-11-29
**Summary of Discussion Phase: Consensus Reached and Additional Results**

Dear Area Chair,

We understand that due to the recent administrative reset, you have been assigned to our paper with the review scores reverted. We realize this creates additional workload for you to reconstruct the discussion progress.

To assist your assessment, we respectfully summarize the **consensus reached during the discussion period**. Specifically, the reviewers who initially gave lower scores mentioned that their main concerns were addressed and stated their intent to raise their ratings before the system reset.

### **1. Resolution of Concerns & Commitment to Raise Scores**

* **Reviewer NLQk (Original Score: 4 -> Raised)**
    * **Reason for upgrade:** The reviewer's primary concern was the comparison between our method and Rejection Sampling (IFQL). After we provided detailed analyses showing our method is structurally superior and achieves significantly higher success rates (51% vs 37%), the reviewer was convinced.
    * **Reviewer's Quote (Nov 22):** *"My main concern relating to the comparison between this method and rejection-sampling was addressed... I still think this is a valuable contribution and have changed my rating."*

* **Reviewer ndGM (Original Score: 2 -> Raised)**
    * **Reason for upgrade:** This reviewer initially questioned the performance gain over baselines like SAC. In our response, we provided new experiments on OGBench and D4RL, demonstrating that our method solves complex tasks where SAC fails completely.
    * **Reviewer's Quote (Nov 17):** *"I thank the authors for the detailed response. In particular the comparison with SAC is important and supports the author's claims... I will raise my final Rating"*.

### **2. Support from Initial Reviewers**

* **Reviewer frzm (Score: 8) & Reviewer hfGV (Score: 8):**
    * Both reviewers maintained their strong acceptance ratings. Reviewer frzm highlighted that our work *"precisely identifies two previously overlooked weaknesses"* and *"consistently outperforms prior state-of-the-art approaches"*. Reviewer hfGV commended the *"clarity of the paper"* and *"exhaustive set of experiments"*.

### **3. Key Additional Experiments & Clarifications**

During the rebuttal, we provided supplementary results to substantiate our claims. We hope you can take these into account:

* **Comparison vs. additional Baselines (Addressing ndGM):** We added a direct comparison against **SAC**. Results show our method significantly outperforms them on complex, multi-modal tasks (e.g., **99% vs 0%** success rate on *Cube Double Play* and **77% vs 0%** on *AntSoccer*).
* **Superiority over Rejection Sampling (Addressing NLQk):** We added an experiment comparing our method against *FQL with Rejection Sampling (FQL-RS)*. Our method achieved a **51%** success rate compared to FQL-RS's **37%**, proving that our learned prior is more effective than simple test-time filtering.
* **Theoretical Analysis (Addressing NLQk & ndGM):** We formalized our approach as an **"Amortized Optimization"** framework to reply reviewer ndGM. We derived a performance bound proving that our amortized policy inherits the gains of best-of-N search while maintaining inference efficiency.
* **Robustness Analysis (Addressing frzm):** We conducted sensitivity analyses on BC weights and entropy coefficients, confirming our method is robust against reward hacking and Q-value overestimation.

We hope this summary helps clarify the status of our submission. We believe that our method represents a significant and verified step forward for efficient flow-matching policies.

Best regards,
The Authors

---

### Meta-Review · Area_Chair_k2fv · 2025-12-28

**Summary:**

1.Argmax over Q-values  may amplify critic bias and reduce diversity, proposed by Reviewer frzm.

2.How the teacher adapts to the distributional shift of this initial distribution, proposed by Reviewer hfGV.

3.Results using the advantage noise selection module only over some tasks in the OGBench, proposed by Reviewer hfGV.

4.Weakness of the theory section, proposed by Reviewer NLQk and Reviewer ndGM.

5.The new method does not show any benefit in the locomotion tasks, also limited performance improvement over FQL, proposed by Reviewer ndGM.

6.Lack of comparisons with SAC as it is a standard exploration approach for continuous control task, proposed by Reviewer ndGM.

**Reviewer Concerns:**

1.Argmax over Q-values  may amplify critic bias and reduce diversity, proposed by Reviewer frzm: during the rebuttal phase, the authors discuss the potential overestimation and reward hacking. It shows that all Q are overestimated, and it indeed encourages reward hacking. They also does a sensitivity analysis of alpha_1, which shows that the Q-guided prior directs the agent to high-value regions. Thus I think this point is addressed by the rebuttal.

2.How the teacher adapts to the distributional shift of this initial distribution, proposed by Reviewer hfGV: during the rebuttal phase, the authors claim that the distributional shift is controlled as the VAE fitting is sufficient from both theoretical and empirical sides. Thus I think this point is addressed by the rebuttal.

3.Results using the advantage noise selection module only over some tasks in the OGBench, proposed by Reviewer hfGV: during the rebuttal phase, the authors clarify that the CVAE is designed to predict a coarse-grained "advantage noise distribution" rather than the action distribution directly. Also, they compare FQL-VAE with FQL-RS(Rejection Sampling), which shows the effectiveness of VAE. Thus I think this point is addressed by the rebuttal.

4.Weakness of the theory section, proposed by Reviewer NLQk and Reviewer ndGM: during the rebuttal phase, the authors claim that the paper does not focus on proposing a new theory, but providing a formal explanation for why the method improves. Thus I think this point is addressed by the rebuttal.

5.The new method does not show any benefit in the locomotion tasks, also limited performance improvement over FQL, proposed by Reviewer ndGM: during the rebuttal phase, the authors claims that achieving stable improvements over a strong baseline is also worth presenting and their method achieved significant gains in other environments. I think it is acceptable, and this point is addressed by the rebuttal.

6.Lack of comparisons with SAC as it is a standard exploration approach for continuous control task, proposed by Reviewer ndGM: during the rebuttal phase, the authors claim that they already compare with RLPD, which is designed to improve SAC for utilizing offline data. They add experimental results of SAC(Offline to Online). Thus I think this point is addressed by the rebuttal.

**Reviewer Scores:**

Reviewer frzm would keep his or her score as 8 if he or she has been able to participate fully in the discussion.

Reviewer hfGV would keep his or her score as 8 if he or she has been able to participate fully in the discussion.

Reviewer NLQk would increase his or her score from 4 to 6 if he or she has been able to participate fully in the discussion.

Reviewer ndGM would increase his or her score from 2 to 4 if he or she has been able to participate fully in the discussion.

---

### Decision · Program_Chairs · 2026-01-26

Accept (Poster)